# *Alterbute*: Editing Intrinsic Attributes of Objects in Images

**Tal Reiss** [1 2]  **Daniel Winter** [1]  **Matan Cohen** [1]  **Alex Rav-Acha** [1]  **Yael Pritch** [1]  **Ariel Shamir** [* 1 3]  **Yedid Hoshen** [* 1 2]

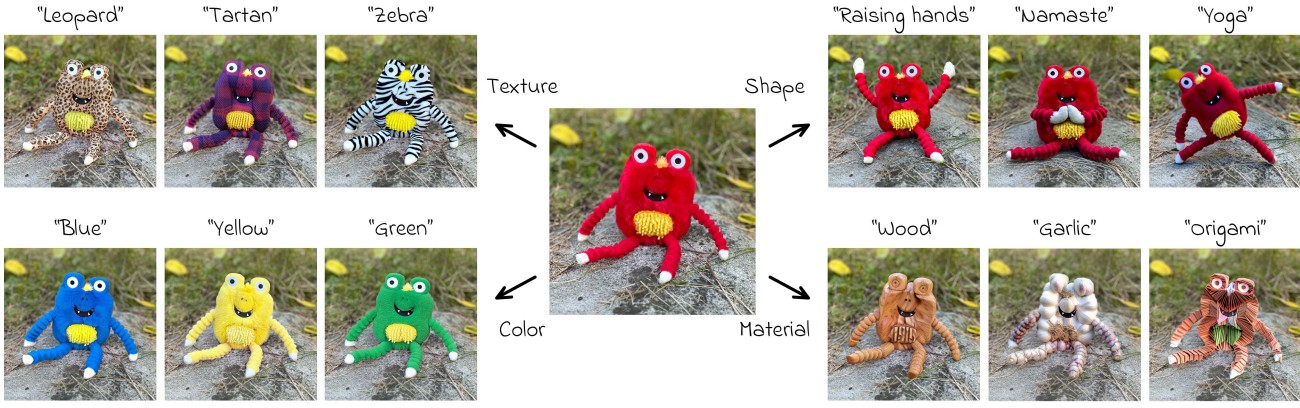

*Figure 1.* Given an input image (center) and a text prompt describing the desired intrinsic attribute, *Alterbute* performs object intrinsic attribute edits, specifically changes to color, texture, material or shape, while faithfully preserving the object's identity.

## Abstract

We introduce *Alterbute*, a diffusion-based method for editing an object's *intrinsic* attributes in an image. We allow changing color, texture, material, and even the shape of an object, while preserving its perceived identity and scene context. Existing approaches either rely on unsupervised priors that often fail to preserve identity or use overly restrictive supervision that prevents meaningful intrinsic variations. Our method relies on: (i) a relaxed training objective that allows the model to change both intrinsic and extrinsic attributes conditioned on an identity reference image, a textual prompt describing the target intrinsic attributes, and a background image and object mask defining the extrinsic context. At inference, we restrict extrinsic changes by reusing the original background and object mask, thereby ensuring that only the desired intrinsic attributes are altered; (ii) *Visual Named Entities* (VNEs) – fine-grained visual identity categories (e.g., "Porsche 911 Carrera") that group objects sharing identity-defining features while allowing variation in intrinsic attributes. We use a vision-language model to automatically extract VNE labels and intrinsic attribute descriptions from a large public image dataset, enabling scalable, identity-preserving supervision. *Alterbute* outperforms existing methods on identity-preserving object intrinsic attribute editing. Our project page is available at https://talreiss.github.io/alterbute/

## 1. Introduction

Editing an object in an image means changing some of its properties while trying to preserve its *identity*. But what defines the identity of an object? An object's appearance in an image is derived from a combination of intrinsic properties: color, texture, material, and shape, as well as extrinsic factors, including camera pose, lighting, and background. Many previous image editing works allow changing extrinsic properties that preserve identity, but few methods can successfully edit *intrinsic* properties. This is the primary challenge we address in this work.

Yet which intrinsic properties are essential to the object's identity, and which can be altered without changing how it is perceived? This choice directly affects the space of feasible, identity-preserving edits. On one extreme, defining identity

---
[*]Equal advising  [1]Google  [2]The Hebrew University of Jerusalem  [3]Reichman University. Correspondence to: Tal Reiss <tal.reiss@mail.huji.ac.il>.

*Proceedings of the 43rd International Conference on Machine Learning*, Seoul, South Korea. PMLR 306, 2026. Copyright 2026 by the author(s).

solely by an object's coarse *category* (e.g., "car") allows for nearly unlimited edits, as long as the result still belongs to the same category. Such a loose definition often conflicts with our intuitive sense of identity. On the other extreme, defining identity at the *instance* level fixes all intrinsic attributes (color, texture, material, shape) and permits almost no variation at all, making meaningful intrinsic edits impossible. Between these two extremes lies a broad spectrum of possible *identity* definitions that trade-off editability against perceived identity preservation.

Most existing editing methods rely on the unsupervised priors of diffusion models and, in practice, only preserve a coarse notion of identity. In contrast, subject-driven generation methods (Ruiz et al., 2023; Chen et al., 2024a; Kumari et al., 2023; Tewel et al., 2023) are conditioned on several views of the object identity but preserve a restrictive notion of identity, disallowing any intrinsic changes.

In this paper, we introduce *Alterbute*, a diffusion-based method for editing intrinsic object attributes while preserving its perceived identity. Following the bitter lesson in machine learning (Sutton, 2019) and recent results in image editing (Lee et al., 2025; Magar et al., 2025), we opt for a supervised approach. However, directly tackling this task ideally requires many image pairs depicting the *same* scene and object, differing only in intrinsic attributes. Such data are virtually nonexistent and challenging to gather or create.

Our first technical innovation is to relax the original task objective: instead of developing a method that exclusively edits intrinsic attributes, we train a model capable of editing both intrinsic and extrinsic attributes. Specifically, we condition the model on three inputs: (i) an identity reference image that captures the object's identity, (ii) a textual prompt describing the target intrinsic attributes, and (iii) a background image and object mask that define the extrinsic context of the target scene. At inference, we constrain the model to preserve extrinsic context by reusing the original background and mask, ensuring that only the intrinsic attributes are altered. While this may appear to overgeneralize the task, it introduces a critical advantage: object image pairs with both intrinsic and extrinsic changes are far easier to find than those with only intrinsic changes. This makes it feasible to tackle the general task in a supervised manner, which is infeasible in the original task.

This relaxation raises a second challenge: how to define identity in a way that allows both intrinsic and extrinsic changes while remaining faithful to human perception. Defining identity using the distance of semantic features (e.g., DINOv2 (Caron et al., 2021; Oquab et al., 2023)) is often too coarse, grouping together perceptually different object identities. In contrast, instance-retrieval features (Cao et al., 2020; Shao & Cui, 2022) result in overly restrictive identities, which do not allow any changes to intrinsic at-

tributes. Therefore, as our second innovation, we propose a middle ground: we define the identity of the object by its *Visual Named Entity* (VNE). VNEs are visual identity categories (e.g., "Porsche 911 Carrera", "IKEA LACK table", "iPhone 16 Pro") that align with the way people naturally refer to specific object types. VNEs group visually similar instances that share a common name, allowing for variation in intrinsic attributes while preserving perceived identity.

We extract VNEs using a large vision-language model (Gemini (Team et al., 2023)) applied to a large public dataset based solely on visual appearance, filtering out generic or unnameable objects. This process yields clusters of images where each cluster corresponds to a single VNE and contains objects that share identity under our new definition but vary in both intrinsic and extrinsic attributes. We further prompt Gemini to describe the intrinsic properties of each VNE object, producing the attribute-level text prompts used during training. This fully automated pipeline enables scalable and identity-consistent supervision without manual labeling.

At inference time, given a source image and a text prompt specifying the change of the target attribute (e.g., "color: red", "material: wood"), *Alterbute* modifies only the specified intrinsic attribute while preserving all other intrinsic and extrinsic properties, as well as the identity of the object. *Alterbute* supports edits across all intrinsic attributes using a *single* unified model, achieving state-of-the-art results on challenging intrinsic object attribute edits.

*Our main contributions are:*

- A method for editing any intrinsic object attribute using a single model, while preserving identity.

- The use of *VNE*s as an effective representation of object identity for intrinsic object attribute image editing, and a VLM-based method for extracting them.

- A relaxed training objective that learns from both intrinsic and extrinsic edits during training, while constraining inference to intrinsic attributes, enabling supervised learning without rare intrinsic-only data.

**Conflict of Interest Disclosure.** The authors are employed by Google, which develops Gemini. Gemini was used as one of three VLM-based evaluation judges in this work (alongside GPT-4o and Claude) and for automated data curation. It was not used for model training or inference.

## 2. Related Works

Diffusion models have become the standard for high-fidelity text-to-image generation (Rombach et al., 2022; Ramesh et al., 2022). Extensions such as inpainting (Yang et al.,

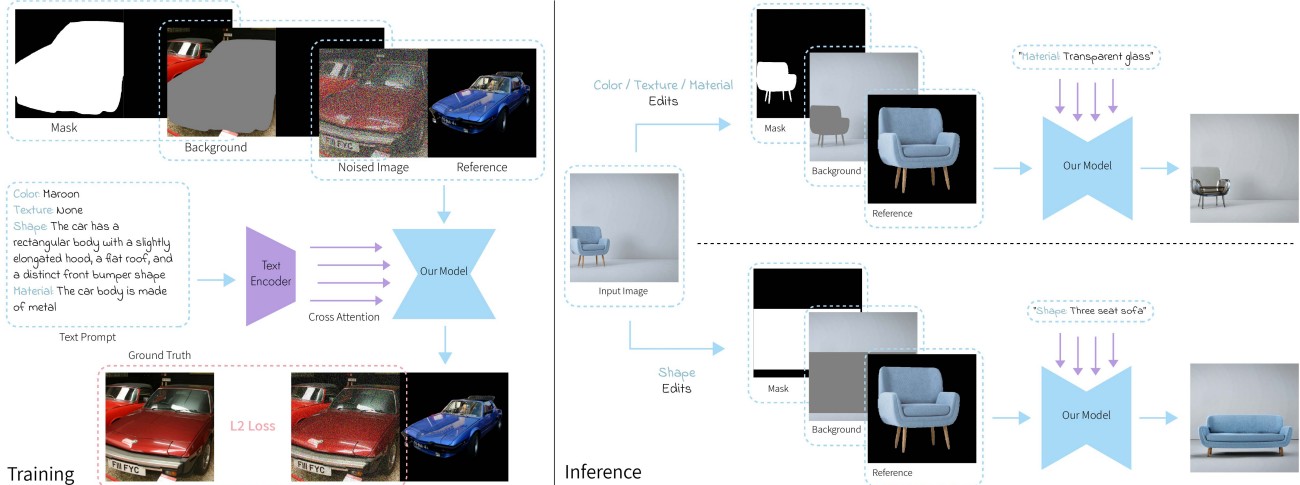

*Figure 2.* **Overview of *Alterbute*.** *Alterbute* fine-tunes a diffusion model for text-guided intrinsic attribute editing. *Left (Training):* Inputs are arranged in a $1 \times 2$ image grid. The left half contains the noisy latent of the target image, while the right half contains a reference image sampled from the same VNE cluster. The model is conditioned on this reference image, a textual prompt describing the desired intrinsic attributes, a background image, and a binary object mask (both represented as grids). The diffusion loss is applied only to the left half to focus the learning on the edited region. *Right (Inference):* Using the same architecture (grid omitted for clarity), *Alterbute* edits the input image directly by reusing its original background and mask. For color, texture, or material edits, we use precise segmentation masks (top). For shape edits where the target geometry is unknown, we use coarse bounding-box masks (bottom).

2023), style transfer (Hertz et al., 2024; Frenkel et al., 2024), and image-to-image (Ye et al., 2023) have broadened their applicability. Instruction-based methods (Brooks et al., 2023; Kawar et al., 2023; Sheynin et al., 2024; Hu et al., 2024; Zhao et al., 2024) offer prompt-driven edits but struggle with intrinsic attribute edits. Other works target spatial or prompt-based manipulation (Yang et al., 2023; Song et al., 2023; 2024; Hertz et al., 2022; Meng et al., 2021), but they are not designed to edit intrinsic object attributes. In contrast, *Alterbute* enables intrinsic attribute editing while preserving object identity and scene context.

**Identity preservation and object personalization.** Preserving object identity during image editing remains challenging. Personalization methods (Ruiz et al., 2023; Gal et al., 2023; Arar et al., 2024) enable high-fidelity generation but require per-object optimization and do not support intrinsic edits. Inversion-based methods (Mokady et al., 2023; Voynov et al., 2023; Garibi et al., 2024) optimize latent codes to reconstruct input images but struggle with identity preservation during editing. More recently, tuning-free approaches (Li et al., 2023; Wei et al., 2023; Xiao et al., 2024; Shi et al., 2024; Ma et al., 2024; Chen et al., 2024c) have been proposed to preserve identity without test-time optimization. (Winter et al., 2025) introduces a grid-based self-attention mechanism conditioned on instance-retrieval-based references, enabling object insertion. However, it does not support intrinsic changes. Diptych (Shin et al., 2025), based on FLUX (Labs, 2024), supports reference-based generation via input grids but struggles with intrinsic

edits. In contrast, *Alterbute* is tuning-free, built on a simpler backbone (SDXL (Podell et al., 2023)), and uses VNEs for identity-preserving attribute editing.

**Intrinsic attribute manipulation.** Several recent methods focus on editing specific intrinsic object properties (Richardson et al., 2024; Cheng et al., 2024). (Sharma et al., 2024) enables changes to physical properties such as albedo and roughness using synthetic data. (Garifullin et al., 2025) enables zero-shot material transfer from reference images, and (Chen et al., 2024b) allows stylized texture editing. (Chen et al., 2023) guides texture synthesis using textual prompts. Although effective within their respective domains, these methods are limited in scope: they target a single attribute type, and often fail to maintain identity and scene consistency. In contrast, *Alterbute* supports editing all intrinsic attributes using a single unified model, while preserving both object identity and scene context.

## 3. Method

Our goal is to edit an object's intrinsic attributes – color, texture, material, or shape – while preserving its identity and maintaining all extrinsic scene properties. To achieve this, we propose a diffusion-based approach whose overview is presented in Fig. 2.

### 3.1. Problem formulation

We consider an input image $y$ depicting a physical 3D scene $s$, and an object $o$ with identity $id$. The goal is to edit $o$

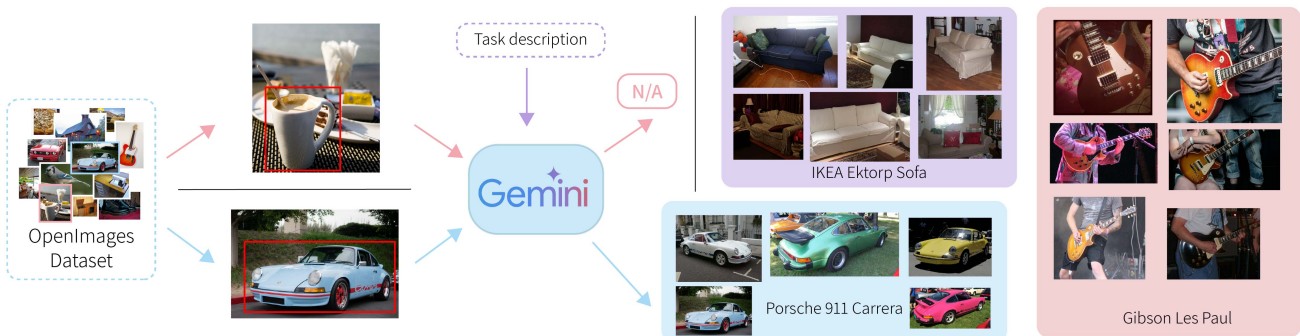

*Figure 3.* We use Gemini to assign textual VNE labels to objects detected in OpenImages. VNE objects (e.g., "Porsche 911 Carrera") are grouped into VNE clusters, while unlabeled instances are filtered out. Example clusters are shown on the right. For each VNE-labeled object, we additionally prompt Gemini to extract intrinsic attribute descriptions, which serve as textual prompts $p$ during training.

according to a textual prompt $p$ that specifies the desired intrinsic attributes (e.g., material: "wood"), while preserving the object's identity and the surrounding scene. Let $G_{\text{physics}}$ denote the physical image formation process that renders the scene. The image can be expressed as:

$$y = G_{\text{physics}}(o, s) \tag{1}$$

We define the object's appearance as a function of its identity $id$, intrinsic attributes $a_{\text{int}}$ (color, texture, material, shape), and extrinsic scene factors $s$ (e.g., camera pose, illumination, and background):

$$o = O(id, a_{\text{int}}, s) \tag{2}$$

The editing operation aims to generate an output image $y'$ in which the object has new intrinsic attributes $a'_{\text{int}}$, guided by the prompt $p$, while keeping both its identity and the extrinsic factors unchanged:

$$y' = G_{\text{physics}}(O(id, a'_{\text{int}}, s), s) \tag{3}$$

While $G_{\text{physics}}$ is conceptually well-defined, the formulation in Eq. (3) is not directly usable for editing real images, as it assumes full knowledge of the physical scene and object geometry, information that is typically unavailable. Therefore, the core challenge becomes learning a generative model that approximates $G_{\text{physics}}$ and supports controlled, identity-preserving object intrinsic attribute editing.

### 3.2. Relaxed training objective

Training a supervised model to edit only intrinsic attributes while keeping extrinsic factors fixed is highly challenging. A naive approach would require paired images of the *same* object with varying intrinsic attributes but identical extrinsic context, such data do not occur naturally and are difficult to collect. To overcome this challenge, we relax the training objective: instead of restricting the model to intrinsic edits alone, we allow it to modify both intrinsic and extrinsic attributes during training. Paired examples of this broader

setting are easier to obtain, making supervised training feasible, unlike the original restricted task.

To enable identity preservation within this relaxed setup, the model is conditioned on three inputs: (i) a reference image that captures the object's identity; (ii) a textual prompt specifying the target intrinsic attributes; and (iii) a background image and binary mask that define the extrinsic scene context and the object's spatial location. The training objective is to generate an image $y'$ in which the object from $id$ appears with the attributes specified in $p$, and is seamlessly composited into $bg$ at the location defined by $m$.

### 3.3. Visual named entities for identity conditioning

For identity conditioning, we define an object's identity through its *Visual Named Entity* (VNE). VNEs are fine-grained visual identity categories (e.g., "Porsche 911 Carrera", "iPhone 16 Pro") that reflect how people naturally refer to specific object types. Unlike broad categories (e.g., "car"), which are too coarse and permit excessive variation that conflicts with our intuitive sense of identity, or instance-level identifiers, which are overly restrictive and allow minimal variation, VNEs strike a practical balance. Specifically, VNEs group visually similar objects sharing a common semantic label, permitting variations in intrinsic and extrinsic attributes while preserving identity. An ablation study in Sec. 4.3 highlights the critical role of VNEs in enabling effective identity-preserving attribute editing.

To extract VNEs at scale, we leverage the OpenImages dataset (Kuznetsova et al., 2020) along with Gemini (Team et al., 2023). For each object detected in OpenImages, Gemini is prompted to assign a VNE label based on the visual characteristics of the object. This process yields clusters of images in which all objects share the same VNE label but exhibit natural variations in their intrinsic (as well as extrinsic) attributes. These clusters serve as the basis for generating training triplets of the form (identity reference, attribute prompt, background + mask), as described in Sec. 3.2.

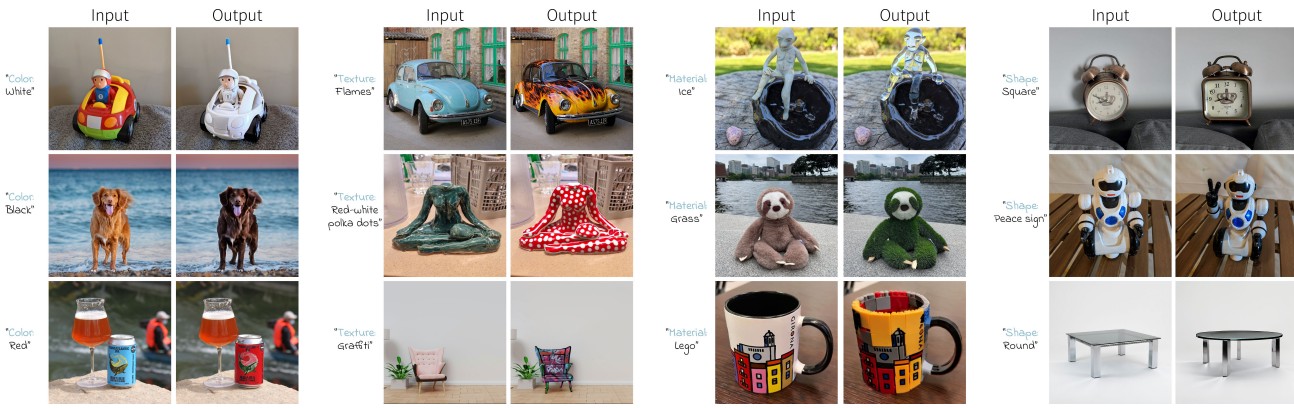

*Figure 4.* **Qualitative results across intrinsic editing tasks.** *Alterbute* successfully edits a variety of intrinsic attributes.

This automatic curation pipeline allows our method to scale across thousands of distinct identities without requiring any manual labeling. Example clusters are shown in Fig. 3.

After forming the VNE clusters, we annotate each object with its intrinsic attributes. To do this, we prompt Gemini to describe each VNE object based solely on its visual appearance, extracting intrinsic properties, specifically color, texture, material, and shape. Gemini returns a structured text output, formatted as key-value pairs for each intrinsic attribute. An example of this output is shown in Fig. 2 (left) and further detailed in App. F. These attribute-level descriptions serve as the textual prompt $p$ used during training.

### 3.4. Training and model architecture

We fine-tune a pretrained latent diffusion model (Ramesh et al., 2022; Rombach et al., 2022; Saharia et al., 2022) to enable precise control over an object identity, intrinsic attributes, and extrinsic scene context. Specifically, we adapt the UNet-based denoising network $D_\theta$ to condition on three inputs: (i) a reference image $id$ containing only the foreground object (with the background masked out), used for identity conditioning. This image is randomly sampled from the same VNE cluster as the target image; (ii) a textual prompt $p$ describing the desired intrinsic attributes; (iii) a scene description $s = (bg, m)$, where $bg$ is a background image and $m$ is a binary mask indicating the object's target location. The network $D_\theta$ learns to map these inputs to the denoised target image $y'$. Training is performed using the standard diffusion L2 loss:

$$\mathcal{L}(\theta) = \mathop{\mathbb{E}}_{\substack{\tau \sim U([0,T]) \\ \epsilon \sim \mathcal{N}(0,1)}} \left[ \sum_{i=1}^{N} \| D_\theta(\alpha_\tau y_i + \sigma_\tau \epsilon, id, p_i, s_i, \tau) - \epsilon \|^2 \right]$$

(4)

where $\tau$ denotes the diffusion timestep, and $\alpha_\tau$, $\sigma_\tau$ parameterize the noise schedule.

To enable identity conditioning, we organize the inputs into a $1 \times 2$ image grid, each with a resolution of $512 \times 512$,

resulting in a composite input image of size $512 \times 1024$. The left half contains the noisy latent of the target object, and the right half contains the reference object image $id$, sampled from the same VNE cluster as the input image. This spatial layout allows self-attention layers in the UNet to propagate identity features across the two halves. The model computes the loss only over the left half, ensuring that denoising is focused exclusively on the target region.

In addition, we provide the model with a background image $bg$, in which the object region is masked with gray pixels, and a binary mask $m$ that specifies the object's target location. These inputs are placed only in the left half of the grid; the right half is filled with zeros. All inputs are concatenated along the channel axis. For the reference image $id$, we always mask out the background to avoid scene leakage and ensure that identity conditioning is purely object-focused (see Fig. 2). The text prompt $p$ is encoded using a text encoder and injected into the UNet through cross-attention layers. To support object reshaping, we randomly alternate between using precise segmentation masks and coarse bounding-box masks for $bg$ and $m$ during training. This encourages generalization across mask granularities and enables reshaping when the target mask is unknown (as it will differ from that of the input object).

### 3.5. Inference-time intrinsic attribute editing

At inference, the model edits intrinsic attributes while preserving all extrinsic factors. Given an input image $y$ containing an object $o$ and prompt $p$ specifying a single intrinsic attribute, we: (i) extract the object mask $m$ using a pretrained segmentation model; (ii) crop $o$ and mask its background to form the reference image $id$; and (iii) mask the object region in $y$ with gray pixels to create the background image $bg$. Feeding $id$, $p$, $bg$, and $m$ into the model produces an output where only the specified intrinsic attribute is modified. See Fig. 2 (right) for illustration.

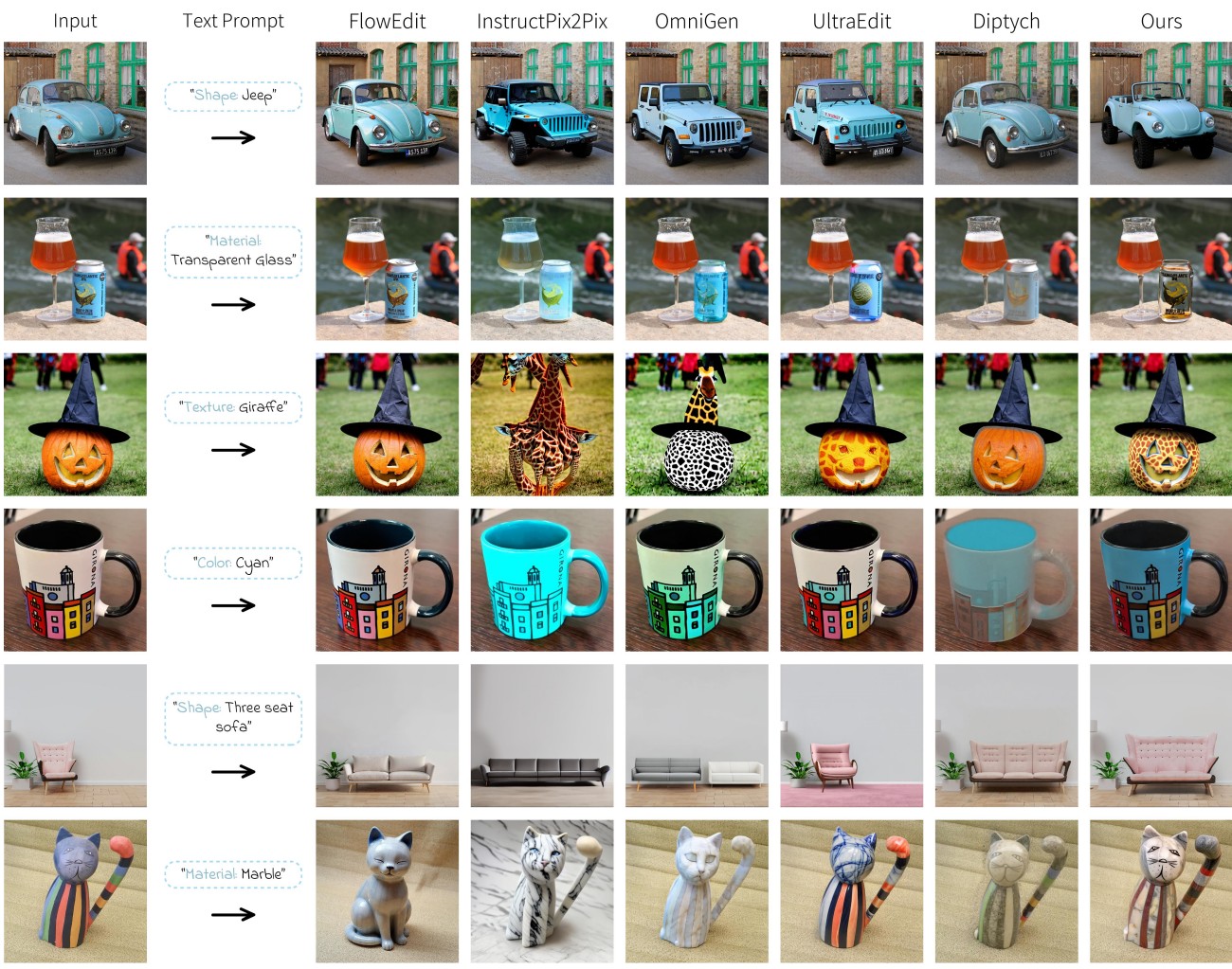

*Figure 5.* **Qualitative comparison.** Baselines often fail to apply the desired edit or preserve identity. In contrast, *Alterbute* produces edits that faithfully reflect the target attribute while maintaining object identity.

## 4. Experiments

We first present qualitative results in Fig. 4, demonstrating that *Alterbute* edits the target attribute while preserving object identity and scene context.

### 4.1. Experimental setting

**Implementation details.** We fine-tune a text-to-image latent diffusion model based on the SDXL architecture (Podell et al., 2023) (with 7B parameters). Segmentation masks are obtained using (Ravi et al., 2024). Our model is trained for $100,000$ steps with a learning rate of $10^{-5}$ and a batch size of 128, using image grids at a resolution of $512 \times 1024$. Training took approximately 24 hours on 128 v4 TPUs. The identity reference $id$ is sampled from the same VNE cluster as the target image. To improve robustness, we randomly drop $id$ while keeping the scene condition in $10\%$ of samples and the text prompt in another $10\%$. Following (Brooks

et al., 2023), we use CFG (Ho & Salimans, 2022) with scales of 7.5 (text) and 2.0 (image). VNE labels and attribute descriptions are extracted using Gemini 2.0 Flash.

**Evaluation data.** As no standard benchmark exists for the task of object intrinsic attribute editing, we construct a dedicated evaluation set comprising 30 distinct objects. Of these, 15 are popular objects commonly used in prior literature, primarily sourced from (Ruiz et al., 2023) and (Gal et al., 2023). To improve diversity, the remaining 15 objects are selected from underrepresented categories such as furniture and vehicles. Each object is paired with multiple text prompts describing different intrinsic attribute modifications, yielding 100 total evaluation samples.

### 4.2. Comparisons

**Baselines.** We compare against both general-purpose and attribute-specific editors: FlowEdit (Kulikov et al., 2025),

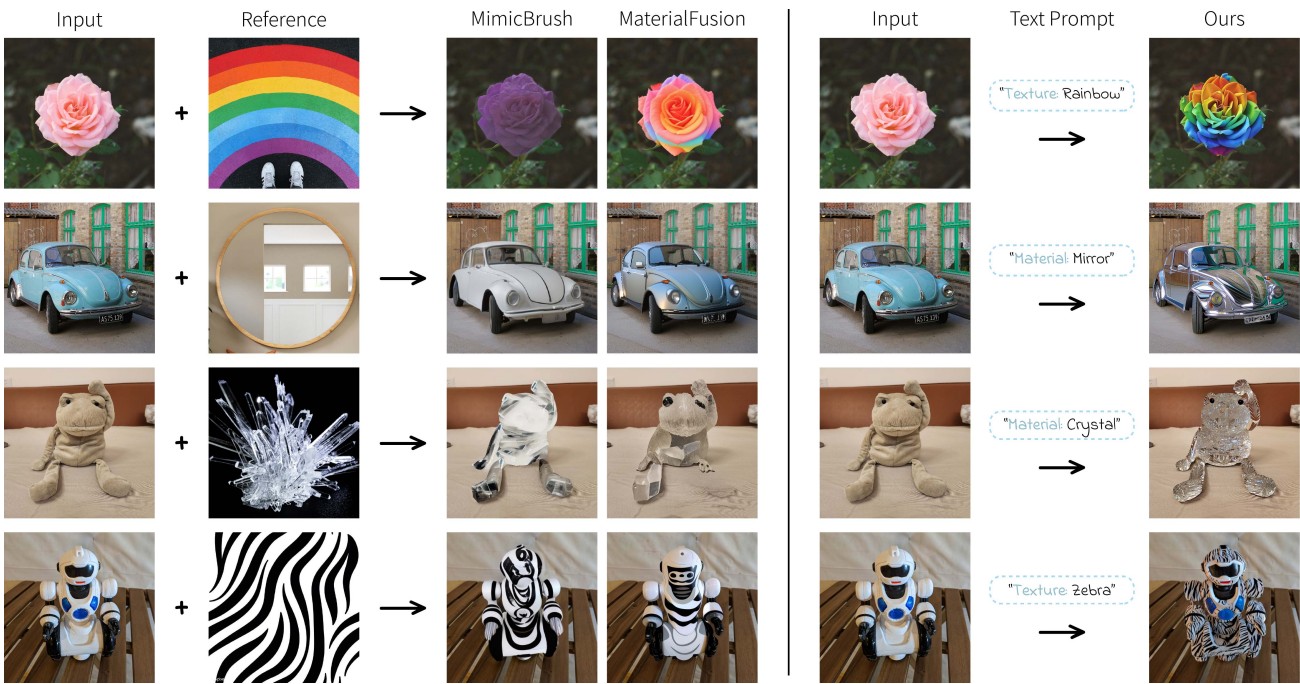

*Figure 6.* **Comparison with attribute-specific editors.** On the left, for MimicBrush and MaterialFusion, we show the input image, reference image, and their edited output. On the right, we present the result produced by *Alterbute*.

InstructPix2Pix (Brooks et al., 2023), OmniGen (Xiao et al., 2025), UltraEdit (Zhao et al., 2024), Diptych (Shin et al., 2025), as well as MaterialFusion (Garifullin et al., 2025) (material) and MimicBrush (Chen et al., 2024b) (texture). Existing colorization methods target a different problem and lack fine object-level control. There are some proprietary systems that have the ability to perform intrinsic manipulations with comparable results to ours, but they lack published papers. We focus our evaluation on open, academic baselines. See App. A for more details.

**Qualitative evaluation.** Fig. 5 compares *Alterbute* with general-purpose image-to-image editing methods. The left-most columns show the input image and the target intrinsic attribute specified as a text prompt. As shown, *Alterbute* successfully modifies the specified intrinsic attribute while preserving both the object's identity and the surrounding scene. In contrast, other methods often struggle to preserve identity or accurately apply the requested edit. Notably, *Alterbute* is the only method capable of identity-preserving object reshaping. Fig. 6 shows comparisons with attribute-specific editors. *Alterbute* consistently achieves better identity preservation and produces high-quality edits across all intrinsic attribute types. Unlike the attribute-specific methods, which are limited to modifying a single attribute type, *Alterbute* supports editing *any* intrinsic attribute within a single unified model.

**Quantitative evaluation.** We evaluated perceptual quality through a user study on the CloudResearch platform with 166 U.S.-based participants. Each rated 20 samples from our 100-case evaluation set, comparing our results with baselines shown in random order. For each sample, participants were shown two edited images in random order: one from our method and one from a baseline. They were asked the following question: *Which result do you prefer based on the text prompt? Consider whether the changes to the object match the prompt and whether the object still looks similar to the one in the input image.* Each sample received five independent ratings, resulting in 500 total ratings per general-purpose baseline and 410 per attribute-specific method, for a total of 3,320 responses. The aggregated results are shown in Tab. 1, where users consistently preferred our method across all comparisons. See App. B for statistical significance analysis. For further evaluation, we conducted a VLM-based evaluation using Gemini (Team et al., 2023), GPT-4o (Achiam et al., 2023), and Claude 3.7 Sonnet (Anthropic, 2025), applying the same protocol and question as used in the user study. As shown in Tab. 1, these VLMs produce preferences that strongly align with user judgments. Additional details are provided in the App. B.

### 4.3. Analysis & Ablation study

**VNE cluster analysis.** We apply our automated VNE labeling pipeline on OpenImages, which contains approximately 9 million images and 16 million object bounding

*Table 1.* **Preference rates (%).** Percentage of comparisons in which evaluators preferred *Alterbute* over each baseline.

| Evaluator | Attribute-specific Editors | | General Purpose Editors | | | | |
| --- | --- | --- | --- | --- | --- | --- | --- |
| | MimicBrush | MaterialFusion | FlowEdit | InstructPix2Pix | OmniGen | UltraEdit | Diptych |
| User | 85.0% | 79.7% | 89.3% | 85.0% | 81.2% | 80.0% | 76.2% |
| Gemini | 94.3% | 87.0% | 89.6% | 88.8% | 80.2% | 86.0% | 76.8% |
| GPT-4o | 89.8% | 77.6% | 88.6% | 87.0% | 77.4% | 78.6% | 74.8% |
| Claude | 92.6% | 81.3% | 92.6% | 85.4% | 78.8% | 85.6% | 77.8% |

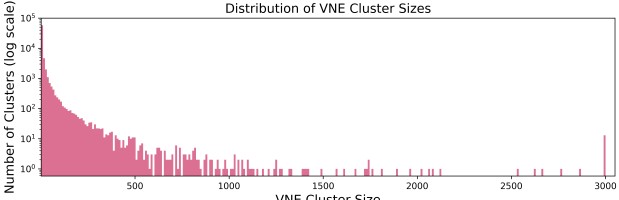

*Figure 7.* Histogram of VNE cluster sizes. The x-axis shows cluster size (number of instances), and the y-axis shows the number of clusters on a log scale. Clusters with more than 3,000 instances are grouped into the last bin.

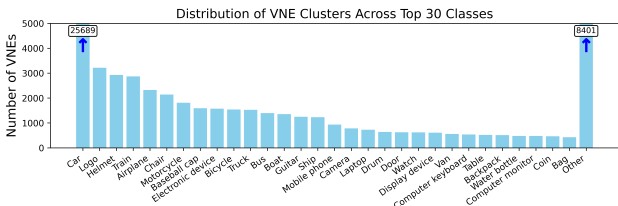

*Figure 8.* Distribution of VNE clusters across the top 30 object classes, sorted by frequency. Each bar shows the number of VNE clusters per class; all remaining classes are grouped into "Other".

boxes. Our pipeline successfully assigns VNE labels to around 1.5 million objects. To ensure sufficient identity supervision, we discard all singleton clusters (i.e., clusters with only one image), resulting in a final set of 69,744 VNE clusters comprising 1,079,442 labeled images. Fig. 7 shows the distribution of cluster sizes, which follows a heavy-tailed pattern: while most clusters are small, a few contain thousands of instances. Fig. 8 presents the distribution of VNEs across the top-30 object classes. A similar long-tailed trend emerges at the category level, with some semantic classes (e.g., "Car") dominating the dataset, while others are sparsely represented. This analysis highlights both the diversity and scale of our curated VNE clusters.

**Grid vs. channel-wise conditioning.** We additionally ablate the conditioning mechanism for the identity reference. Replacing the spatial grid concatenation ($1\times2$ grid enabling cross-image self-attention) with channel-wise concatenation results in no-ops at inference: the model fails to apply the requested attribute edits and outputs the source image largely unchanged. Without cross-image self-attention, the conditioning signal from the identity reference cannot propagate to the target, so the model collapses to the identity mapping. The grid is therefore a structural necessity for fine-grained identity transfer.

**Different identity definitions.** *Alterbute* relies on an identity reference image $id$ sampled from the same VNE cluster as the target object. This reference is crucial for conditioning the model to preserve identity during editing. In this ablation, we investigate how alternative identity definitions

affect the model's performance in intrinsic attribute editing. Specifically, we compare the following strategies for selecting identity references: (i) *DINOv2 feature space:* For each target image, the top-5 most similar objects are retrieved from the OpenImages dataset based on cosine similarity in the DINOv2 (Oquab et al., 2023) feature space. (ii) *Instance-retrieval feature space:* Similar to the previous, but retrieval is based on instance-level retrieval features (Shao & Cui, 2022; Winter et al., 2025). (iii) *In-place editing:* The target image is used as its $id$ reference during training. This setting mirrors inference-time editing. (iv) *Ours:* A reference is sampled from the same VNE cluster as the target image.

While instance-retrieval (IR) features can group visually similar instances, they often fail to provide sufficient variation in intrinsic attributes. DINOv2 performs worse, often clustering visually similar but identity-distinct objects, which damages identity conditioning. Moreover, both DINOv2 and IR operate over the entire OpenImages dataset, often resulting in clusters drawn from semantically unsuitable categories (e.g., food, landmarks, nature), where intrinsic attribute variation is minimal or absent (see App. D for examples). As a result, the training data lacks the diversity needed to learn controlled edits, leading the model to ignore the prompt and overfit to identity and mask information. The in-place editing baseline further emphasizes the importance of our relaxed training formulation: using the same image as both target and reference fails to decouple identity from attributes and cannot generalize to attribute editing. In contrast, our VNE-based strategy ensures identity-preserving yet attribute-diverse supervision. See Fig. 9 for qualitative comparisons of the different identity definitions.

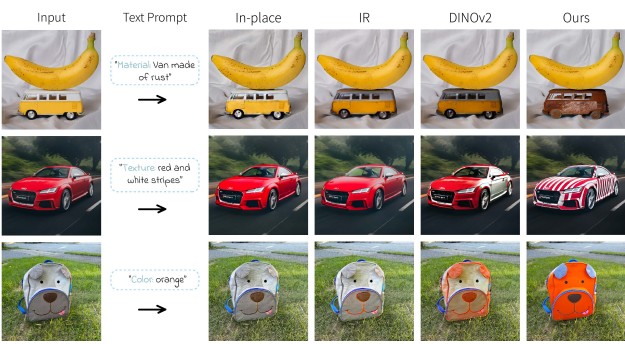

*Figure 9.* **Ablation on identity definitions.** Comparison of identity reference strategies: in-place, DINOv2, IR, and our VNE-based approach. Each row shows the input and target attribute (left), followed by edited results under each identity definition.

## 5. Discussion & Limitations

**Single attribute editing at inference time.** During training, the text prompt $p$ describes all intrinsic attributes of the target object in a key-value format (see Fig. 2 left). However, at inference time, we provide a single key-value text prompt corresponding to the specific intrinsic attribute we wish to edit. Our training enables this flexibility, where we randomly omit the text prompt in $10\%$ of examples, forcing the model to infer unspecified attributes from the reference image. Thus, during inference, *Alterbute* learns to modify only the specified attribute while preserving others, enabling targeted control with minimal text.

**Multi-attribute editing.** Intrinsic attributes often exhibit natural dependencies. For example, changing an object's material to gold implicitly constrains other attributes: it cannot simultaneously have a black color. Our model captures such correlations through the training data and avoids producing contradictory combinations. Nevertheless, *Alterbute* can successfully modify multiple intrinsic attributes simultaneously when those attributes are not in conflict. See the App. C.4 for additional examples and discussion.

**Background artifacts with bounding box masks.** To enable reshaping, *Alterbute* supports coarse bounding box masks instead of precise segmentations. While this improves flexibility, it may cause slight background inconsistencies within the masked region (see Fig. 10 top). A possible remedy is to pre-remove the object, providing a clean background and eliminating the need for masking.

**Reshaping rigid objects.** As shown in Fig. 10 (bottom), reshaping rigid objects does not always yield the desired results. While *Alterbute* supports shape manipulation, editing the geometry of rigid objects remains challenging, as the shape is often correlated with identity-defining features. In some cases, the generated shapes may lack realism or fail to

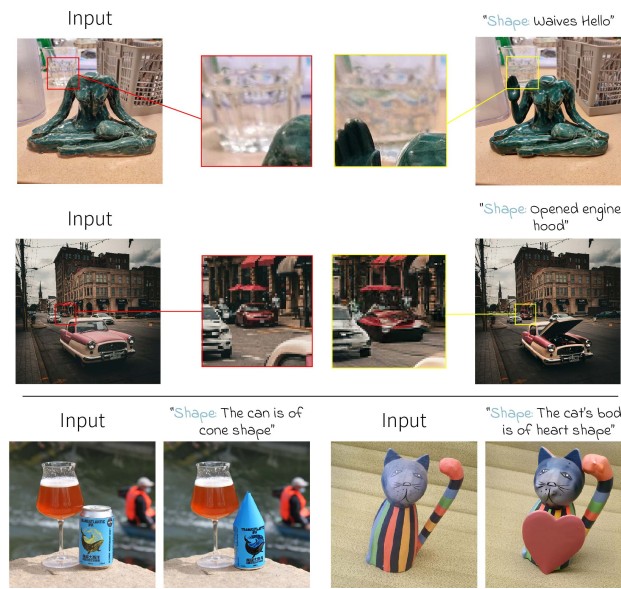

*Figure 10.* **Limitations of *Alterbute*.** *Top:* Background artifacts may occur with coarse bounding box masks. *Bottom:* Shape edits may produce unrealistic or unintended geometries.

reflect the intended change. Nevertheless, *Alterbute* shows promising results despite some limitations.

**VNE annotation bias.** The VNE labeling pipeline relies on Gemini for semantic identity assignment. Categories prominent in Gemini's training distribution may receive better VNE coverage, potentially introducing annotation bias. Our high-confidence filtering mitigates spurious labels, and the pipeline is fully replicable via the prompts in App. F.

**Benchmark scale.** Our evaluation benchmark (30 objects, 100 editing cases) was designed to cover all four attribute types and long-tail categories. While the benchmark captures a diverse range of editing challenges, its scale is limited. Expanding it with additional objects, attribute combinations, and editing scenarios is an important future direction.

## 6. Conclusion

We presented *Alterbute*, a diffusion-based method for editing intrinsic object attributes – color, texture, material, and shape – while preserving object identity and scene context. To address the lack of aligned supervised training data, we introduced a relaxed training objective that learns from both intrinsic and extrinsic changes while constraining inference to intrinsic edits. We also introduced *Visual Named Entities* (VNEs) – visual identity categories extracted automatically using a VLM. VNEs group objects sharing identity-defining features with natural intrinsic variation, making them ideal for supervision. Using a single unified model, *Alterbute* achieves state-of-the-art results in intrinsic attribute editing.

## Acknowledgments

We thank Shira Bar On for creating the figures and visualizations. We also thank Tomer Golany, Dani Lischinski, Yarden Frenkel, Asaf Shul, Shmuel Peleg, Bar Cavia, and Nadav Magar for their valuable feedback and discussions. Tal Reiss is supported by the Google PhD Fellowship.

## Impact Statement

This paper presents a method for editing intrinsic visual attributes of objects in images while preserving their identity. As with any generative image editing technology, there is a potential for misuse in creating misleading visual content. We note that our method exclusively targets non-human objects (e.g., products, furniture, vehicles) and is not designed for, nor trained on, human faces or biometric features, which limits its applicability for identity fraud or deepfake generation. We encourage the community to develop appropriate safeguards as image editing capabilities continue to improve.

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

# A. Baselines

In our qualitative and quantitative comparisons (including a user study), we evaluated several baseline methods that have publicly available open-source models. These baselines fall into two categories: general-purpose image editing models and specialized intrinsic attribute editors. All baselines were run with publicly available code and recommended settings. No baseline was fine-tuned on task-specific data. For stochastic methods, we generated 5 outputs per input and selected the best-quality result, making our reported win rates conservative. All mask-based baselines received the same SAM-extracted mask with morphological dilation (kernel $K=5$) applied identically. Table 2 summarizes the inputs and mask configuration for each baseline.

*Table 2.* Baseline evaluation protocol. Each baseline received the strongest possible setup to ensure conservative win-rate estimates.

| Baseline | Inputs Provided | Mask Used |
|---|---|---|
| FlowEdit | Source image + text | None |
| InstructPix2Pix | Source image + instruction | None |
| OmniGen | Source image + text | Object mask |
| UltraEdit | Source image + instruction | Object mask |
| Diptych | Source image (diptych panel) + text | Object mask |
| MaterialFusion | Source + material reference | Object mask |
| MimicBrush | Source + texture ref region | Target region mask |

We describe each baseline below.

## A.1. General-purpose editors

**FlowEdit** (Kulikov et al., 2025): FlowEdit is a text-driven image editing method that avoids diffusion inversion by using a pretrained flow model to navigate latent space edits. We use the public implementation released by the authors, based on FLUX (Labs, 2024).

**InstructPix2Pix** (Brooks et al., 2023): InstructPix2Pix fine-tunes a diffusion model to follow image editing instructions generated from synthetic instruction-image pairs. It performs text-conditioned edits without per-image optimization.

**OmniGen** (Xiao et al., 2025): OmniGen is a unified diffusion model capable of both text-to-image generation and text-conditioned editing. It handles multiple tasks in a single model via shared reasoning abilities. We use the available checkpoint for evaluation.

**UltraEdit** (Zhao et al., 2024): UltraEdit fine-tunes a diffusion model on a large-scale synthetic instruction-based dataset (approximately 4M examples) for fine-grained editing. It improves precision over previous instruction-following editors. We use the publicly released UltraEdit model trained on the UltraEdit-4M dataset, based on Stable Diffusion 3 (Esser et al., 2024).

**Diptych** (Shin et al., 2025): Diptych uses an inpainting-based approach, framing editing as a two-panel grid (diptych) generation task using pretrained FLUX (Labs, 2024). It enables zero-shot subject-driven edits with strong identity preservation. We use the open-source code.

## A.2. Specialized intrinsic attribute editors

**MaterialFusion** (Garifullin et al., 2025): MaterialFusion performs high-quality, zero-shot material transfer by predicting intrinsic material properties from a single image and applying reference materials. We use the available pretrained model for evaluation.

**MimicBrush** (Chen et al., 2024b): MimicBrush enables zero-shot, localized attribute transfer by imitating visual attributes (texture, material) from a reference image onto a selected target region using diffusion models. The publicly released checkpoints and code are used for our evaluation.

*Table 3.* **Statistical significance of the results.** A binomial statistical test suggests that our results are statistically significant (p-value $< 0.05$)

| Evaluator | Attribute-specific Editors | | General Purpose Editors | | | | |
|---|---|---|---|---|---|---|---|
| | MimicBrush | MaterialFusion | FlowEdit | InstructPix2Pix | OmniGen | UltraEdit | Diptych |
| User | $< 1e-10$ | $< 1e-10$ | $< 1e-10$ | $< 1e-10$ | $< 1e-10$ | $< 1e-10$ | $< 1e-10$ |
| Gemini | $< 1e-12$ | $< 1e-12$ | $< 1e-12$ | $< 1e-12$ | $< 1e-12$ | $< 1e-12$ | $< 1e-12$ |
| GPT-4o | $< 1e-12$ | $< 1e-12$ | $< 1e-12$ | $< 1e-12$ | $< 1e-12$ | $< 1e-12$ | $< 1e-12$ |
| Claude | $< 1e-12$ | $< 1e-12$ | $< 1e-12$ | $< 5e-12$ | $< 1e-12$ | $< 3e-12$ | $< 6e-12$ |

## B. Quantitative Evaluation

**User study.** To evaluate the perceptual quality of our results, we conducted an extensive user study on the CloudResearch platform. A total of 166 participants, primarily based in the United States and selected at random, were recruited for the study. Each participant was shown 20 samples, sampled from our evaluation set of 100 intrinsic attribute editing cases.

For each sample, participants were shown two edited images in random order: one generated by our method and the other by a baseline. They were asked the following question:

> *Which result do you prefer based on the text prompt?*
> *Consider whether the changes to the object match the prompt and whether the object still looks similar to the one in the input image.*

An example of the questionnaire interface is shown in Fig. 11. Each sample received 5 independent ratings, resulting in 500 total ratings for each general-purpose baseline and 410 for each attribute-specific editing method, for a total of 3,320 responses. The aggregated results are reported in Tab. 2 of the main paper. As shown, participants strongly preferred the outputs generated by *Alterbute* over all competing baselines. A binomial test on the collected responses, reported in Tab. 3, confirms that these preferences are statistically significant ($p$-value $< 0.05$).

**VLM-based evaluation.** To complement the user evaluation, we conducted an evaluation using three vision-language models: Gemini (Team et al., 2023), GPT-4o (Achiam et al., 2023), and Claude 3.7 Sonnet (Anthropic, 2025). Each model was asked the same question and shown the same image pairs as in the user study. We collected five ratings per sample using the same evaluation set, yielding 500 and 410 total decisions for general and attribute-specific baselines, respectively, totaling 3,320 comparisons. The aggregated results, reported in Tab. 2, show strong agreement with user preferences. A binomial test (also reported in Tab. 3) confirms the statistical significance of these results.

**Conventional evaluation protocol.** Standard evaluation of image editing models typically considers two axes: identity preservation and alignment with a target textual prompt. To adapt this framework to the task of intrinsic attribute editing, we define the following metrics: (i) Identity preservation is measured using the average pairwise cosine similarity between the edited object and its identity reference image. We report results using two visual feature spaces: DINO and CLIP image embeddings, denoted as DINO and CLIP-I, respectively. (ii) Textual alignment assesses whether the desired intrinsic attribute was successfully applied. This is measured via CLIP image-text similarity (CLIP-T), computed as the cosine similarity between the CLIP embedding of the edited image and the target attribute prompt.

While commonly used, these metrics are not well-suited to intrinsic attribute editing, where the goal is to modify specific attributes without changing the object's identity. For example, methods that fail to apply the edit and simply return the original object may score highly on DINO and CLIP-I despite being ineffective. Additionally, both DINO and CLIP are sensitive to changes in intrinsic attributes, which can result in lower similarity scores even when identity is perceptually preserved. Conversely, methods that generate a new object matching the prompt but discard the original identity can achieve high CLIP-T scores while failing the core objective of identity preservation. In both cases, relying on any single metric provides an incomplete and potentially misleading view of model performance. For completeness, we report results under this conventional protocol in Tab. 4, where *Alterbute* achieves competitive performance and the highest CLIP-T scores. However, we argue that meaningful evaluation of intrinsic attribute editing should jointly assess both identity preservation

*Table 4.* **Standard metrics comparison.** We report identity preservation using DINO and CLIP-I, and target attribute alignment using CLIP-T. Best results are shown in **bold**, and second-best are underlined.

| Method | DINO | CLIP-I | CLIP-T |
|---|---|---|---|
| FlowEdit (Kulikov et al., 2025) | 0.813 | 0.900 | 0.294 |
| InstructPix2Pix (Brooks et al., 2023) | 0.772 | 0.877 | 0.302 |
| OmniGen (Xiao et al., 2025) | 0.823 | 0.912 | 0.305 |
| UltraEdit (Zhao et al., 2024) | **0.841** | **0.922** | 0.303 |
| Diptych (Shin et al., 2025) | 0.794 | 0.901 | 0.313 |
| Ours | 0.815 | 0.914 | **0.321** |

and attribute editing. Our complete evaluation, including qualitative comparisons, user studies, and VLM-based assessments, is tailored to this task and provides a more reliable and task-aligned performance measure.

# C. Additional Results

## C.1. Comparison with Commercial Systems

We additionally compare *Alterbute* against two concurrent commercial systems, FluxKontext and Qwen-image-editing, using the VLM-based evaluation protocol. As shown in Tab. 5, *Alterbute* is on par with Qwen-image-editing and clearly outperforms FluxKontext, confirming it is competitive with state-of-the-art commercial systems despite being a fully transparent academic method.

*Table 5. Alterbute* win rate (%) vs. commercial editing systems.

| Compared to | Gemini | GPT-4o | Claude |
|---|---|---|---|
| FluxKontext | 61.2 | 62.6 | 58.7 |
| Qwen-image-editing | 49.8 | 50.3 | 50.1 |

## C.2. Per-Attribute Breakdown

Tab. 6 reports win rates broken down by attribute type. Performance is consistent across all four attribute categories, with shape edits achieving the highest win rates, likely because shape changes are the most challenging for baselines.

*Table 6.* Per-attribute win rate (%) of *Alterbute* vs. all baselines.

| Attribute | User | Gemini | GPT-4o | Claude |
|---|---|---|---|---|
| Color | 78.6 | 81.2 | 76.4 | 79.3 |
| Texture | 80.1 | 83.7 | 79.3 | 82.3 |
| Material | 82.7 | 87.4 | 83.4 | 86.2 |
| Shape | 88.2 | 91.8 | 88.4 | 91.4 |
| **Overall** | **82.3** | **86.1** | **82.0** | **84.9** |

## C.3. Compute Efficiency

At half the training budget (50K steps), *Alterbute* still wins 75–78% of comparisons against all baselines, a drop of only ∼7 percentage points from the full 100K-step model (Table 7). In a direct head-to-head, VLM evaluators only marginally prefer 100K over 50K (57–61% preference), confirming that halving the budget has far less impact than the quality gap over the baselines.

*Table 7.* Effect of training budget on *Alterbute* win rate (%) vs. all baselines.

|  | Gemini | GPT-4o | Claude |
|---|---|---|---|
| 100K steps vs. baselines | 86.1 | 82.0 | 84.9 |
| 50K steps vs. baselines | 78.0 | 75.7 | 76.3 |
| 100K preferred over 50K | 58.2 | 57.1 | 60.6 |

### C.4. Qualitative Comparisons

We provide extended qualitative comparisons to complement those presented in the main paper. Figs. 12 to 14 show additional results comparing our method to general-purpose image-and-text-to-image editing baselines. Figs. 15 and 16 present further comparisons against specialized attribute-specific editing methods.

As discussed in the main paper, intrinsic attributes often exhibit natural dependencies. Our model captures such correlations through its training data and avoids generating contradictory combinations. At the same time, it supports multi-attribute editing when the requested changes are semantically compatible. Fig. 20 showcases examples where *Alterbute* successfully modifies multiple intrinsic attributes simultaneously, while preserving object identity and maintaining scene context. These results demonstrate the model's capacity to generalize beyond single-attribute at-a-time edits and perform coherent multi-attribute transformations.

## D. Identity Definitions

In the main manuscript, we presented an ablation study comparing several strategies for selecting identity references during training: (i) *DINOv2 feature space:* For each target image, the top-5 most similar objects are retrieved from the OpenImages dataset based on cosine similarity in the DINOv2 (Oquab et al., 2023) feature space. (ii) *Instance-retrieval feature space:* Similar to the previous approach, but based on instance-level retrieval features (Shao & Cui, 2022; Winter et al., 2025). (iii) *Ours:* A reference image sampled from the same VNE cluster as the target object.

Fig. 9 in the main paper shows that identity references selected via DINOv2 or IR fail to support effective intrinsic attribute editing. To further support the analysis presented in Sec. 4.3 of the main paper, we visualize example clusters obtained using each identity definition. Fig. 17 and Fig. 18 display clusters derived from DINOv2 and instance-retrieval features, respectively, with each row corresponding to a single cluster. Although instance-retrieval features can group visually similar objects, they often lack sufficient variation in intrinsic attributes. DINOv2 performs worse, frequently clustering objects that are visually similar yet identity-distinct, which harms identity conditioning. Additionally, because both methods operate over the full OpenImages dataset, they often form clusters from semantically unsuitable categories (e.g., food, landmarks, nature), where intrinsic attribute variation is limited or irrelevant. Consequently, training on such clusters leads the model to ignore the textual prompt and overfit to the identity and mask alone, resulting in poor editing performance. In contrast, our VNE-based clustering ensures that identity references are both identity-consistent and diverse in intrinsic attributes. Fig. 19 shows example clusters obtained with our VNE-based definition.

## E. Editing Both Intrinsic and Extrinsic Attributes

While the primary focus of our work is on editing *intrinsic* object attributes, our relaxed training objective enables the model to handle modifications to both intrinsic and *extrinsic* factors. In this extended setting, the model is given a new background image, a reference object image, and a textual prompt describing the desired attribute modifications. It then composes the object into the new scene while applying the specified intrinsic edits. If the prompt is empty (i.e., " "), the model is instructed to preserve all intrinsic attributes from the reference identity and perform only the insertion of the object into the new context. Fig. 21 showcases examples of such joint intrinsic-extrinsic edits. These results open future directions for applying our approach in broader generative tasks such as subject-driven generation or object insertion with controlled transformations.

## F. Gemini-Based Labeling Pipeline

We use Gemini 2.0 Flash to automatically label both VNEs and the intrinsic attributes of objects. The prompts used for VNE labeling and intrinsic attribute description are shown in Figs. 22 and 23, respectively. For VNE labeling, Gemini returns a label along with a confidence score: {*Low*, *Medium*, *High*}. We keep only examples with a *High* confidence score to ensure label quality and discard any unlabeled or low-confidence instances. In Fig. 24, we show examples of objects along with their corresponding intrinsic attribute descriptions generated by Gemini. We note that recent advances in VLMs make our pipeline feasible, enabling scalable data-driven training that was previously impractical in image editing research.

## G. Best Practices for Inference

**Text prompts.** During sampling, the text prompt controls which intrinsic attribute to modify and what target value to apply. Prompts should follow the key-value format `<attribute>: <value>` (e.g., `color: red`, `material: marble`, `texture: knitted`, `shape: cylindrical`). This mirrors the training-time prompt structure described in Sec. 3. To edit a single attribute while preserving all others, provide only one key-value pair; the model will infer the remaining attributes from the identity reference image (enabled by the 10% prompt dropout during training). Multiple attributes can be edited simultaneously by specifying several key-value pairs, provided the requested changes are semantically compatible (see App. C.4).

**Object masks.** The object mask defines the spatial region where the edit is applied. For standard attribute edits (color, texture, material), SAM automatically extracts the object mask from the source image given a point or bounding box prompt. For reshaping edits, the target shape is unknown a priori, so the user provides a bounding box specifying the desired target region. This bounding box can also be generated automatically using text-based object detectors (e.g., Grounding DINO) given a text description of the object. In all cases, we apply morphological dilation with kernel $K=5$ to the mask to avoid tight boundary artifacts.

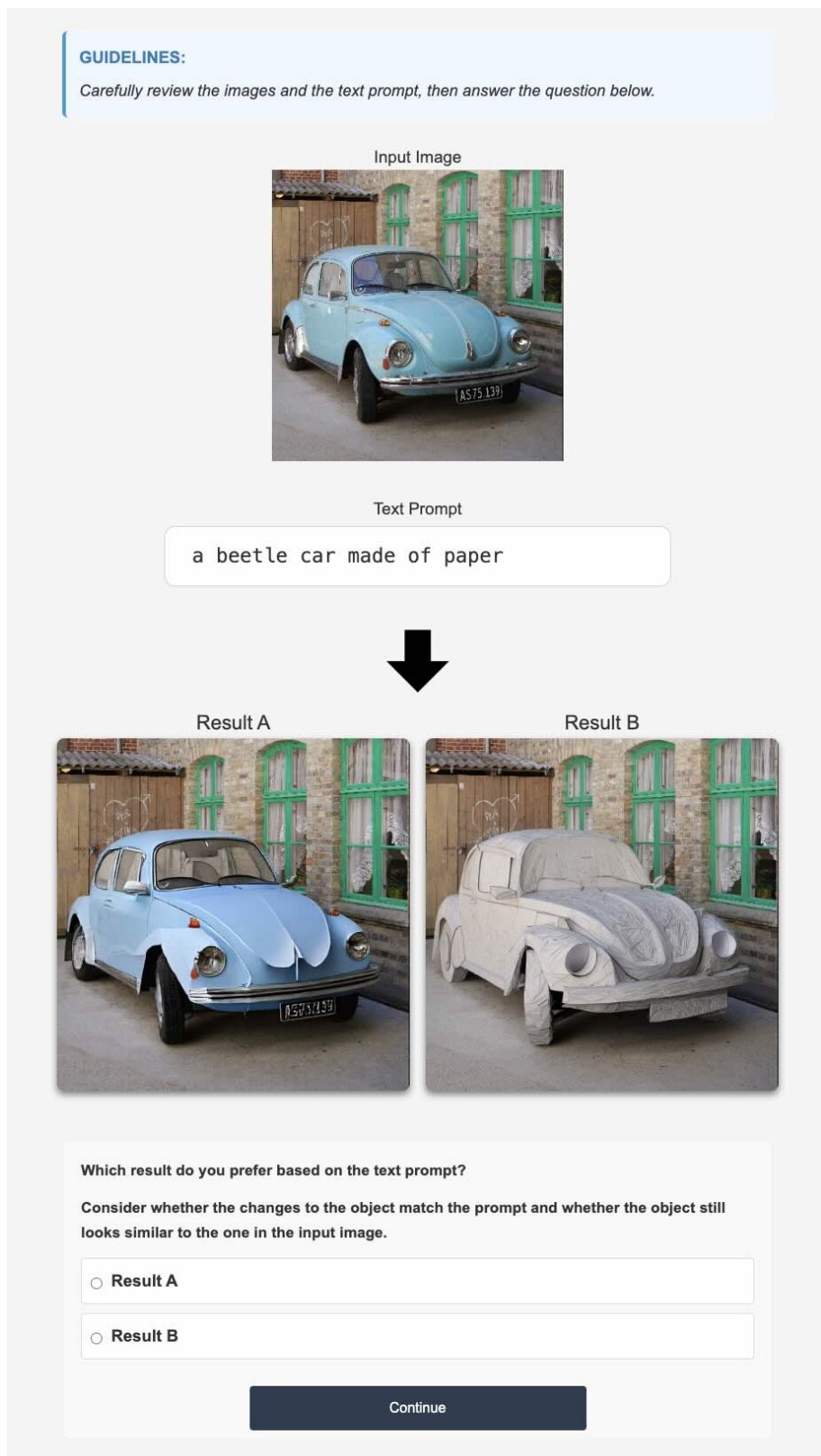

*Figure 11.* A screenshot of the user study questionnaire.

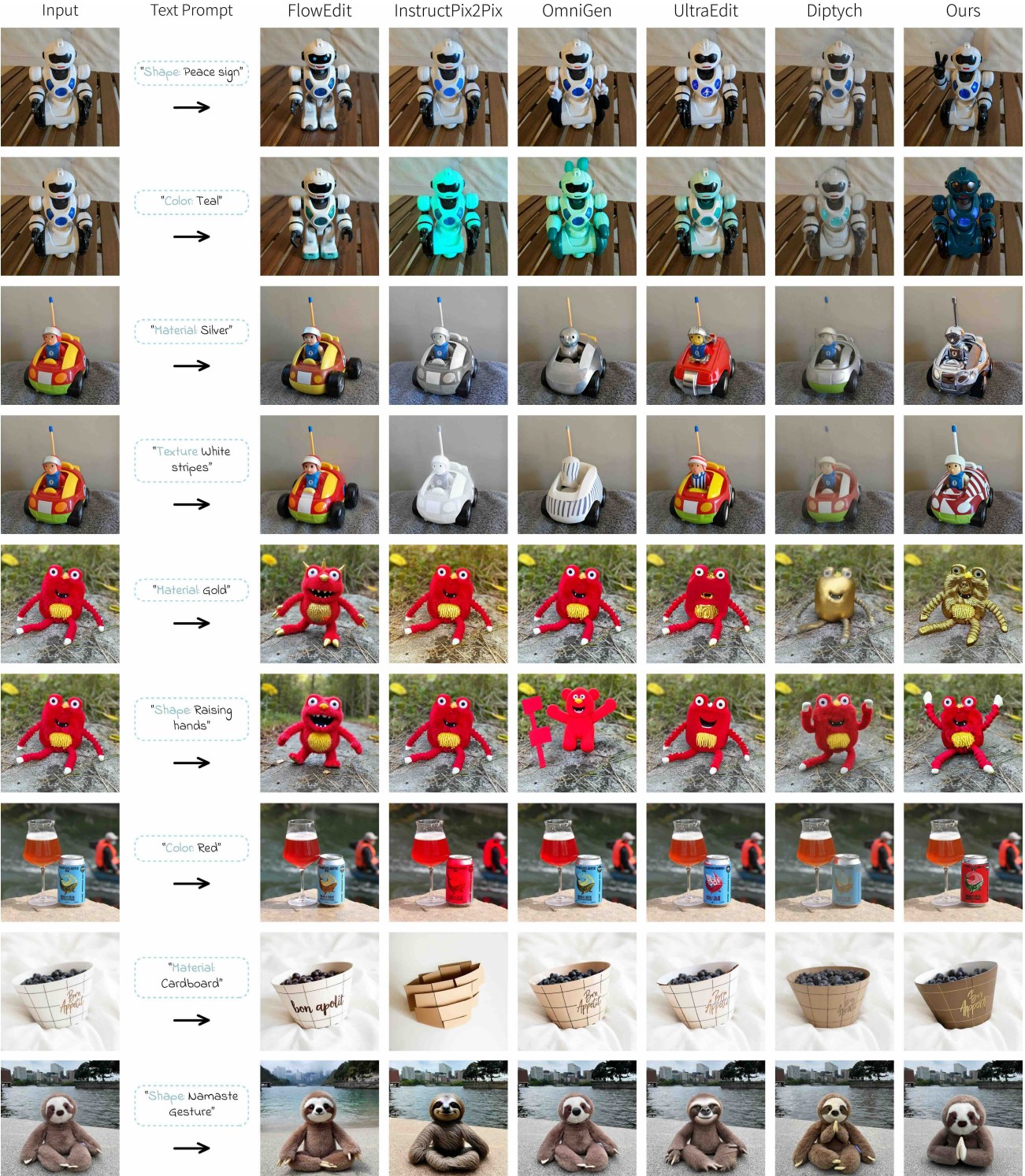

*Figure 12.* Additional qualitative comparisons against general-purpose image-and-text-to-image editing methods.

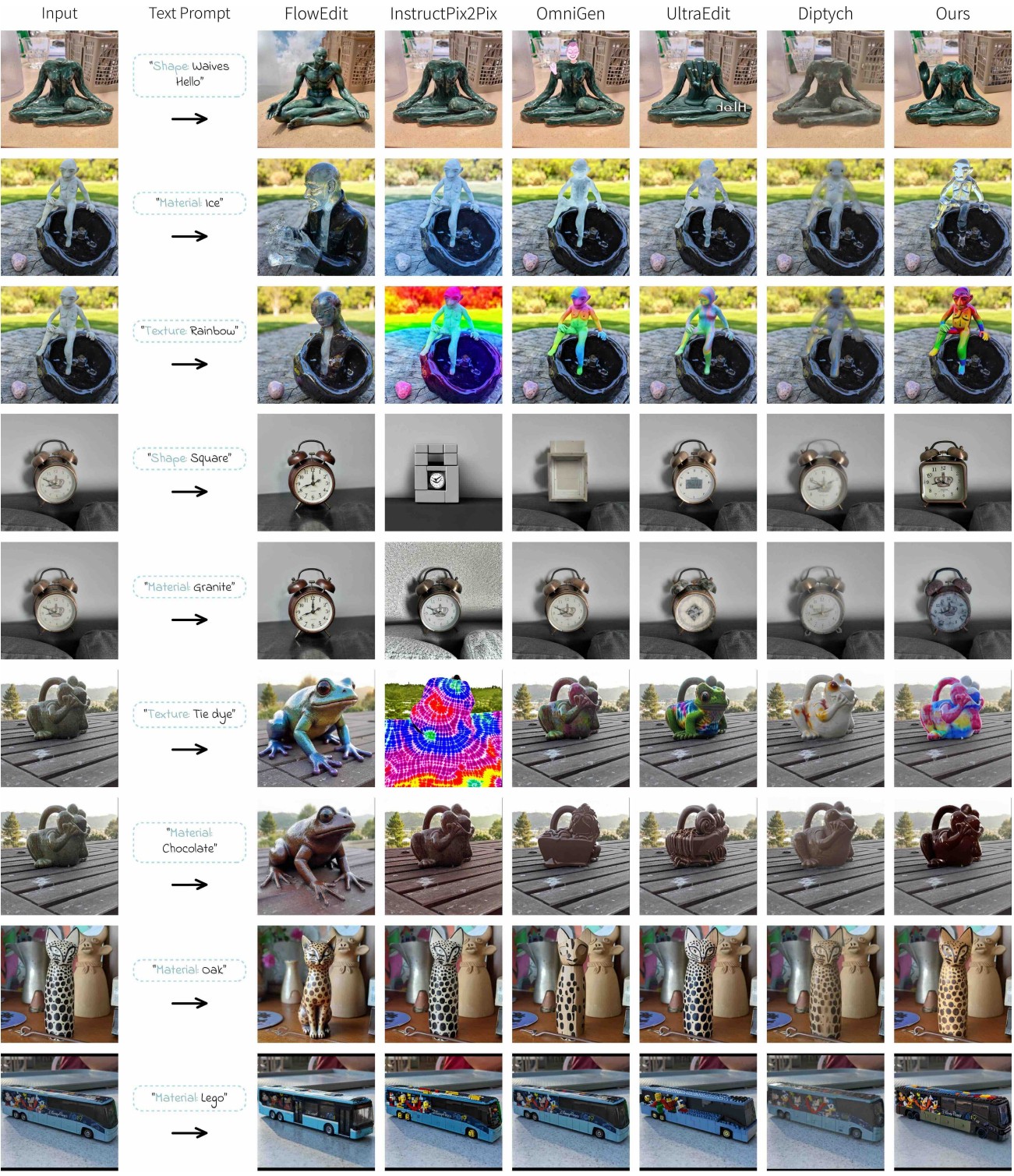

*Figure 13.* Additional qualitative comparisons against general-purpose image-and-text-to-image editing methods.

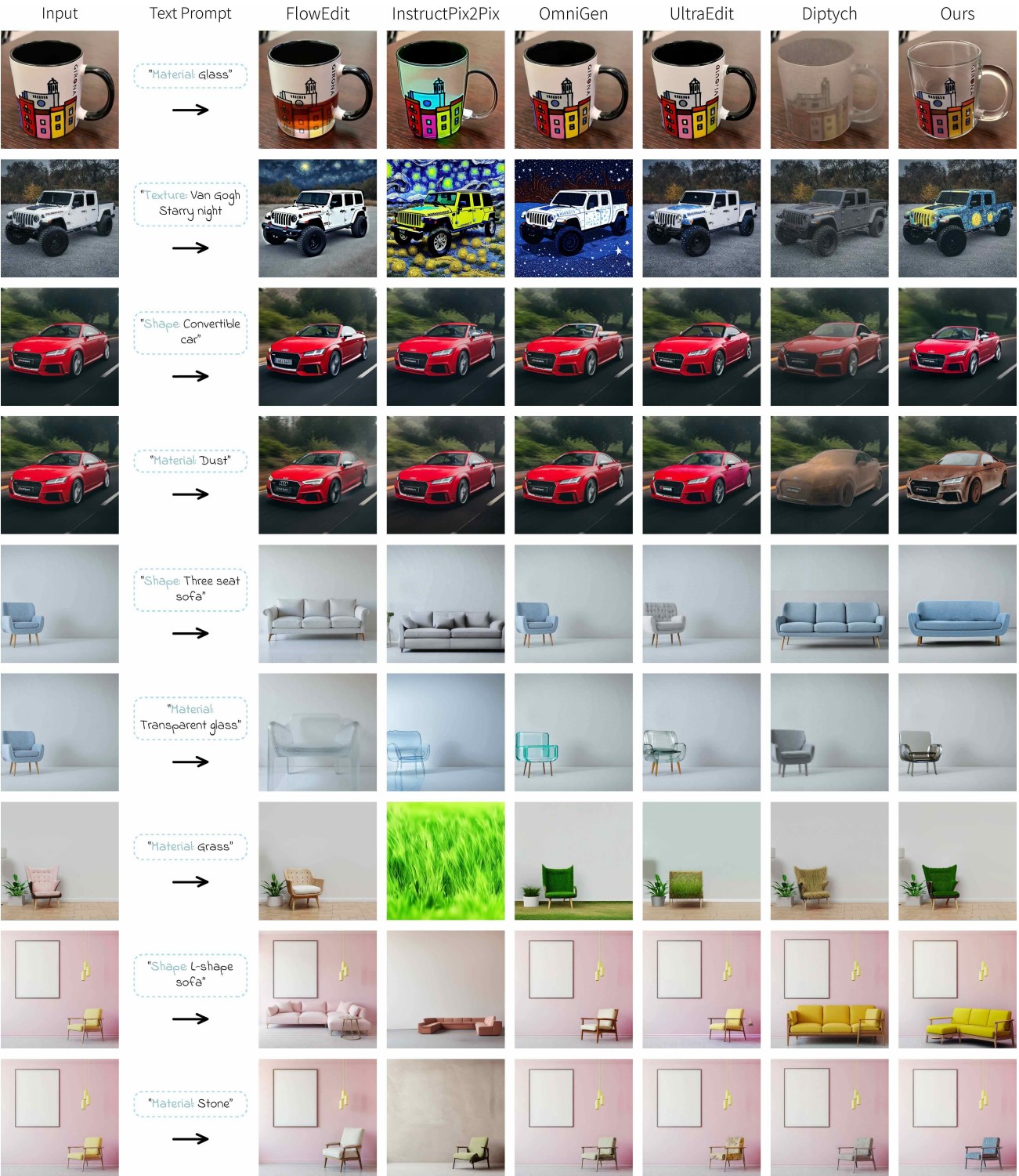

*Figure 14.* Additional qualitative comparisons against general-purpose image-and-text-to-image editing methods.

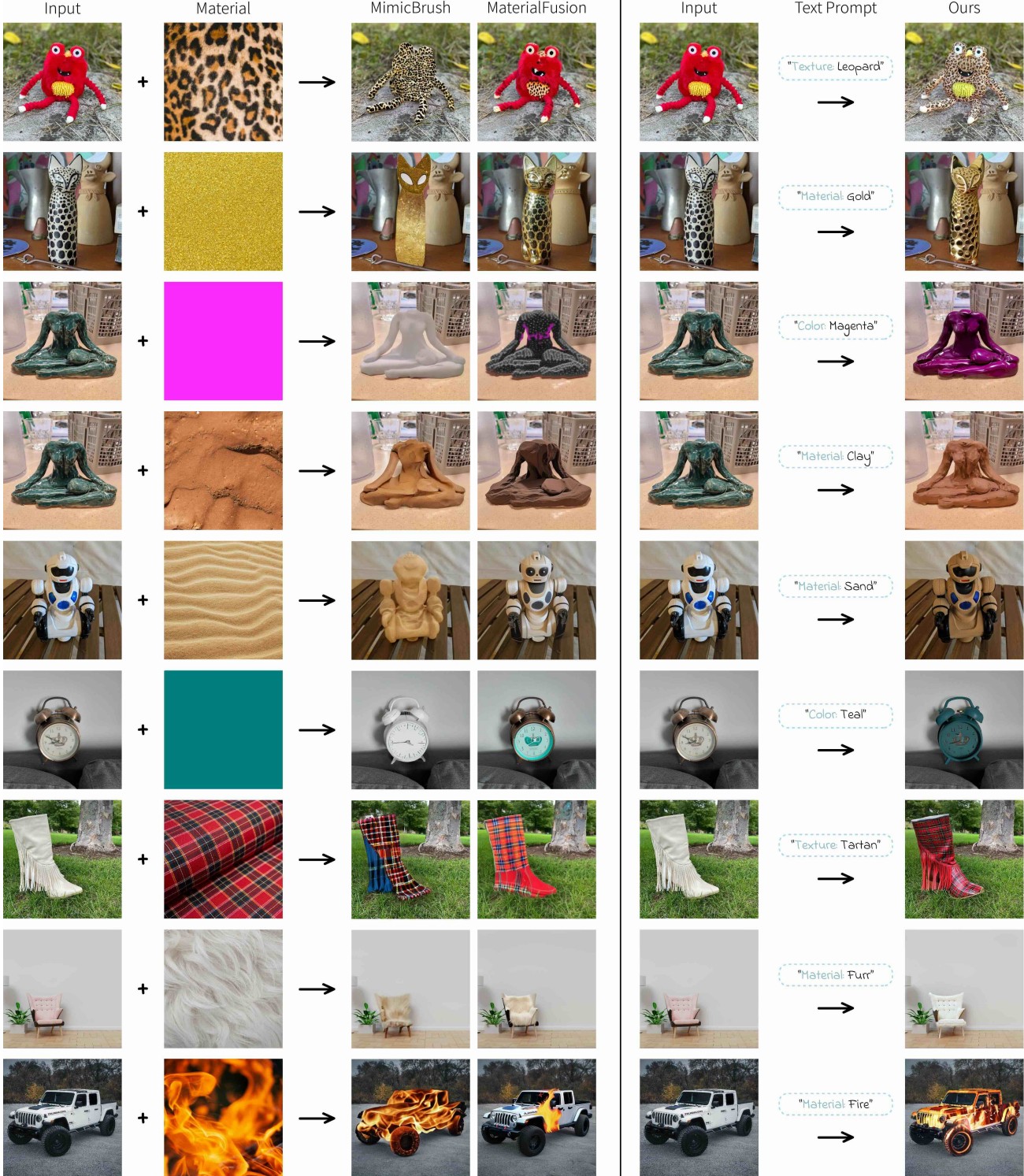

*Figure 15.* Additional qualitative comparisons against attribute-specific edits.

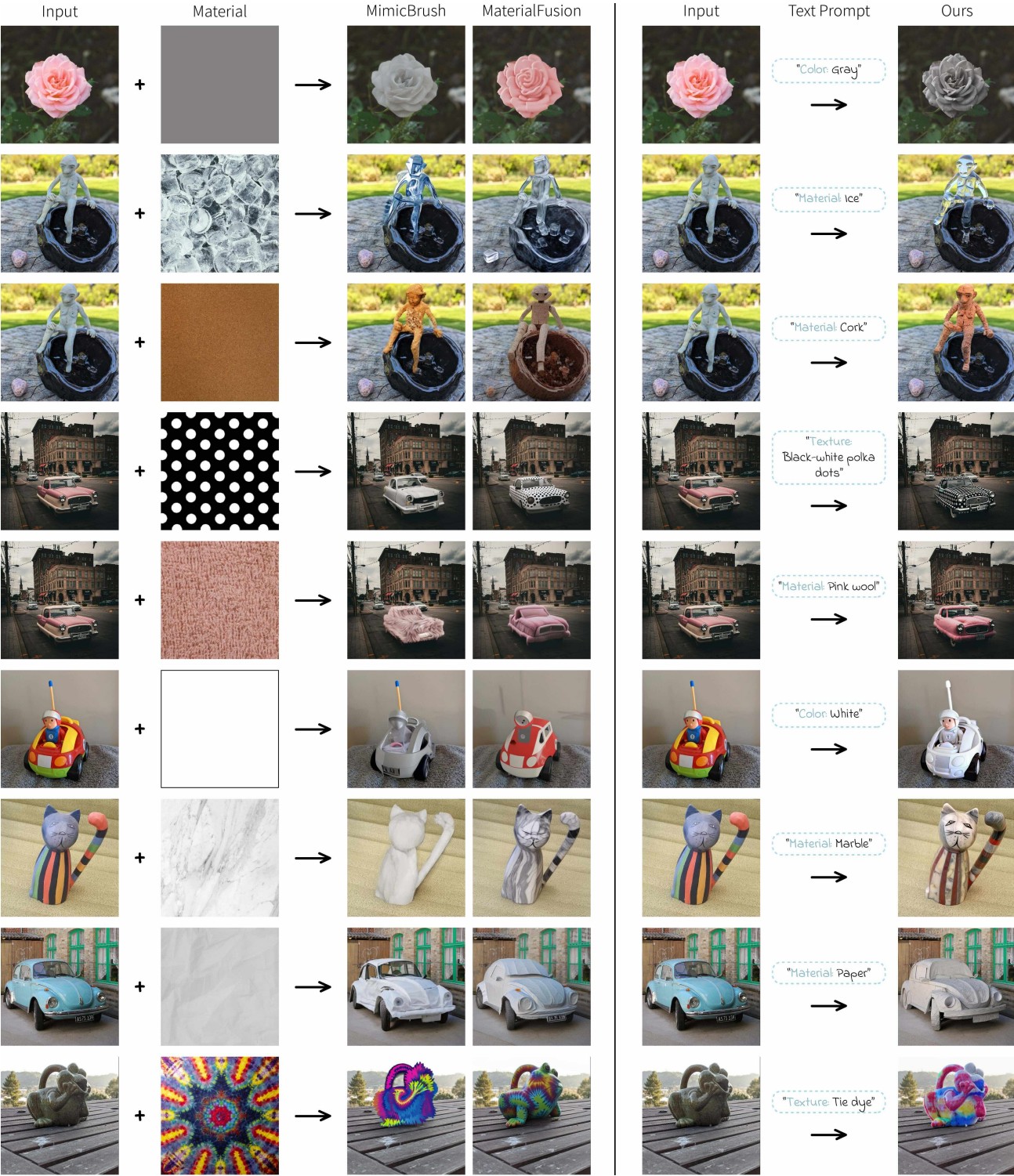

*Figure 16.* Additional qualitative comparisons against attribute-specific edits.

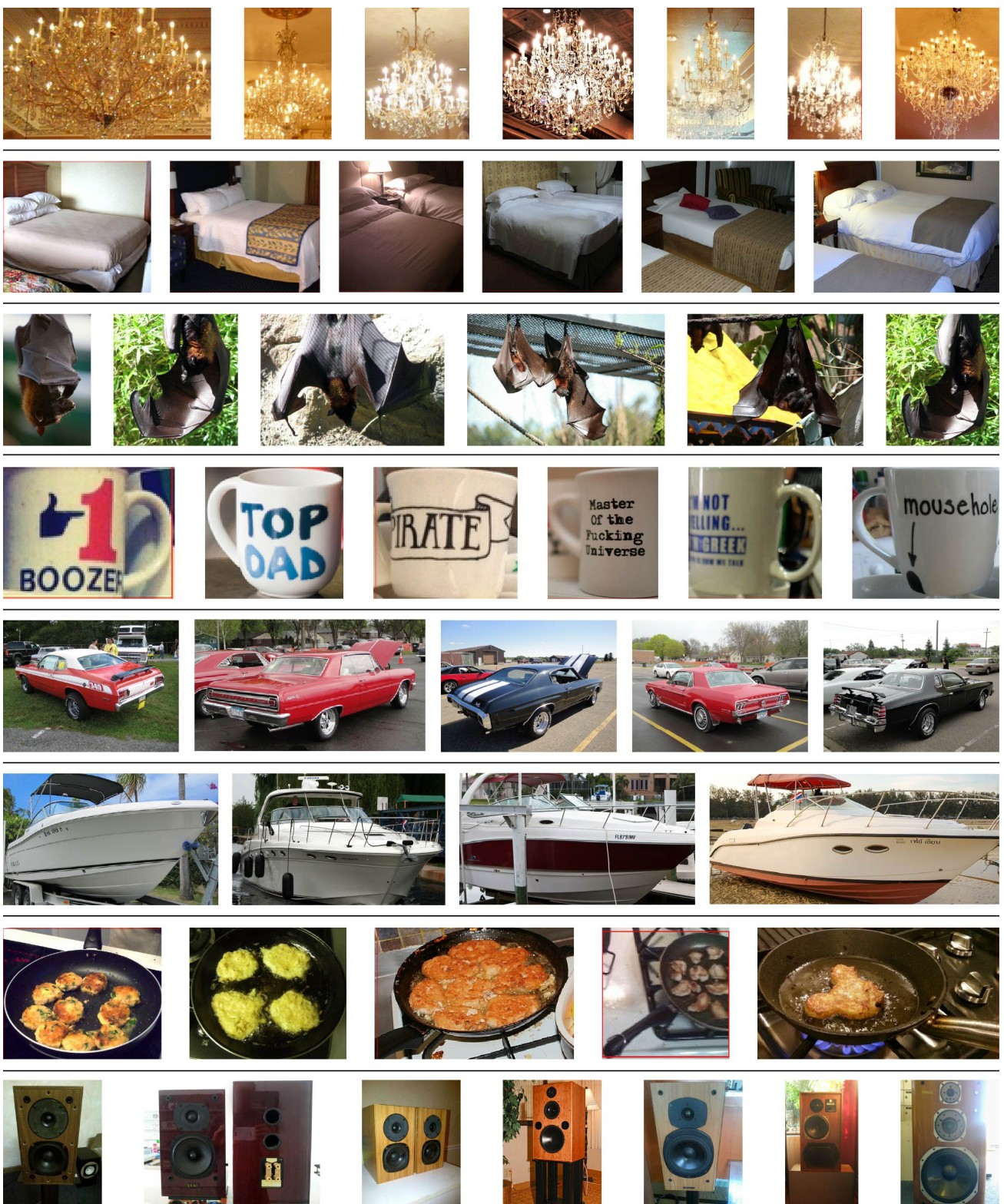

*Figure 17.* Example clusters retrieved using DINOv2 feature similarity, with each row showing images from the same cluster.

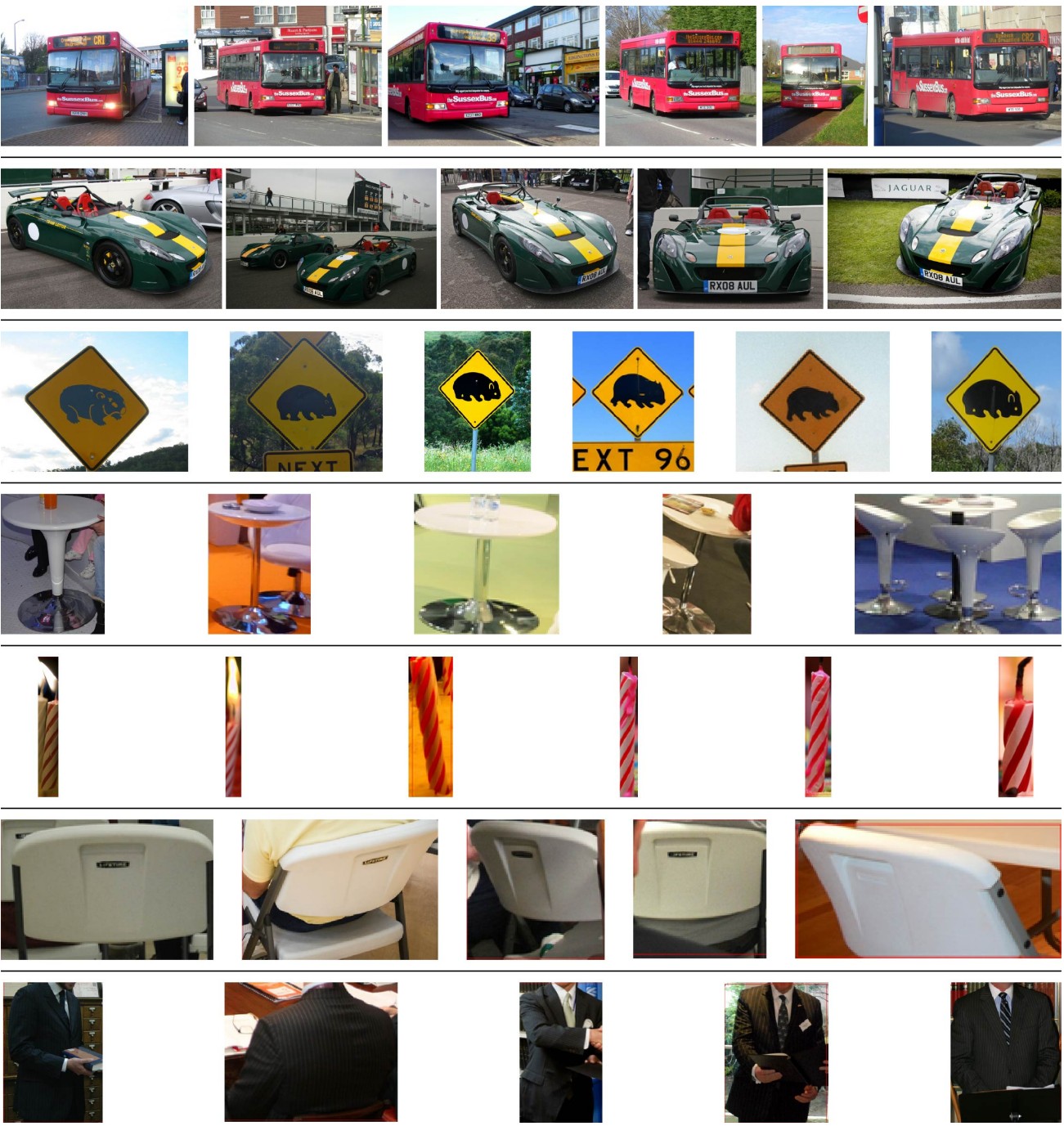

*Figure 18.* Example clusters retrieved using IR feature similarity, with each row showing images from the same cluster.

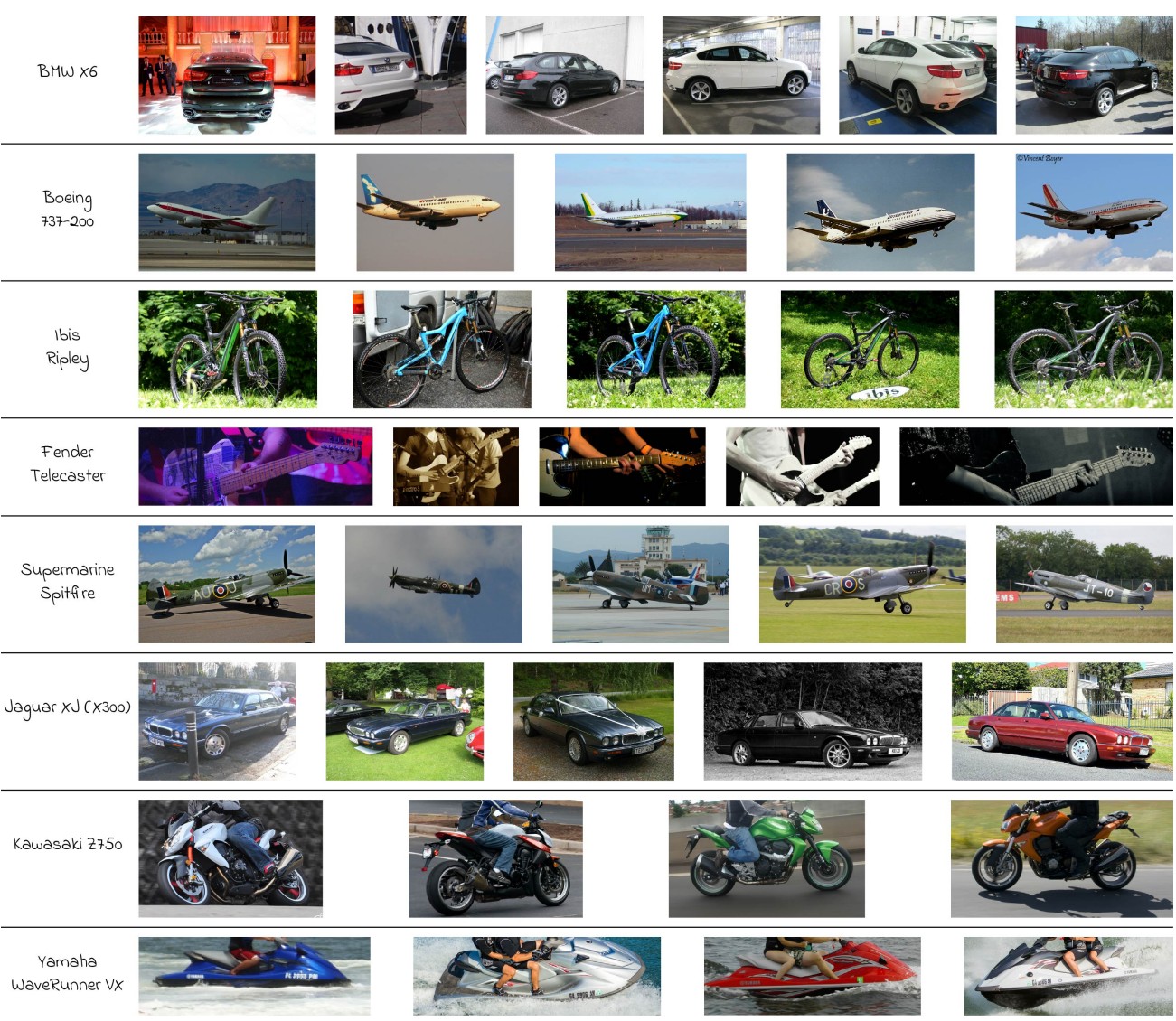

*Figure 19.* Example of VNE clusters.

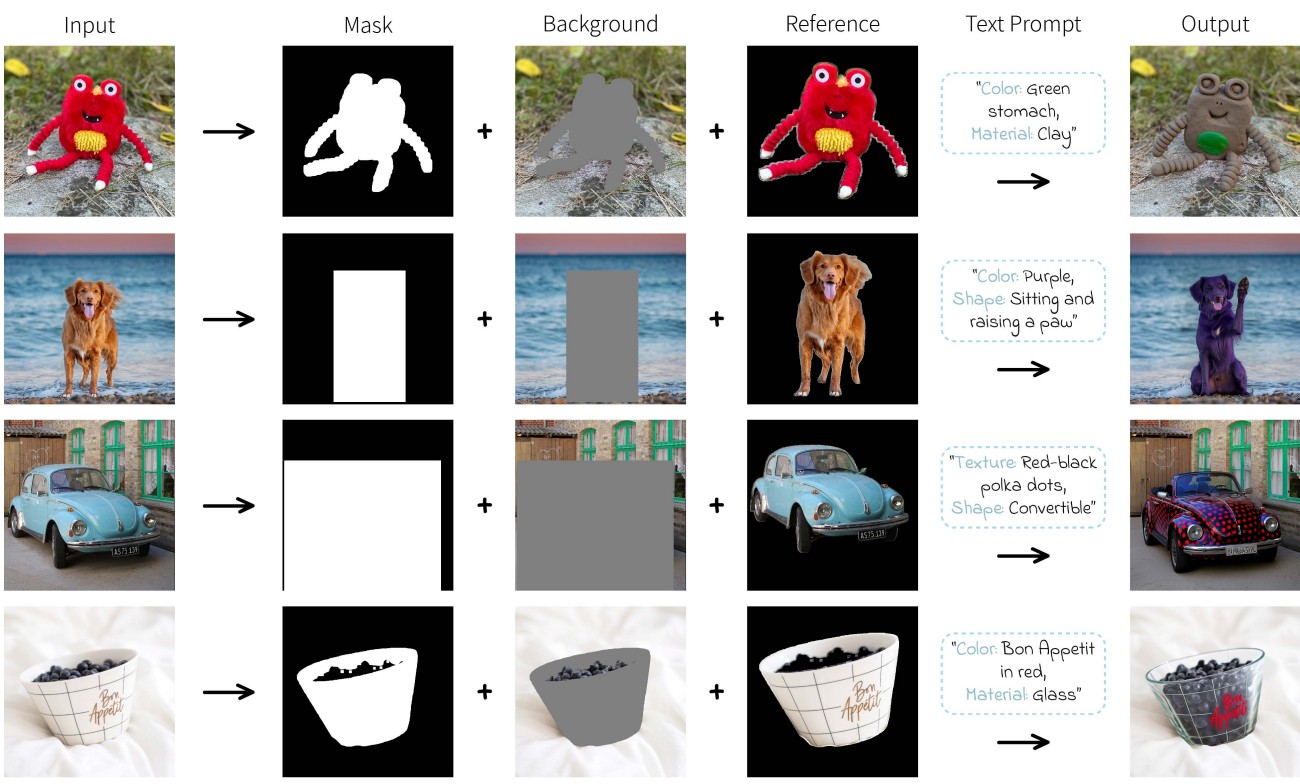

*Figure 20. Alterbute* results on multi-attribute editing, showing its ability to apply compatible edits while preserving identity and context.

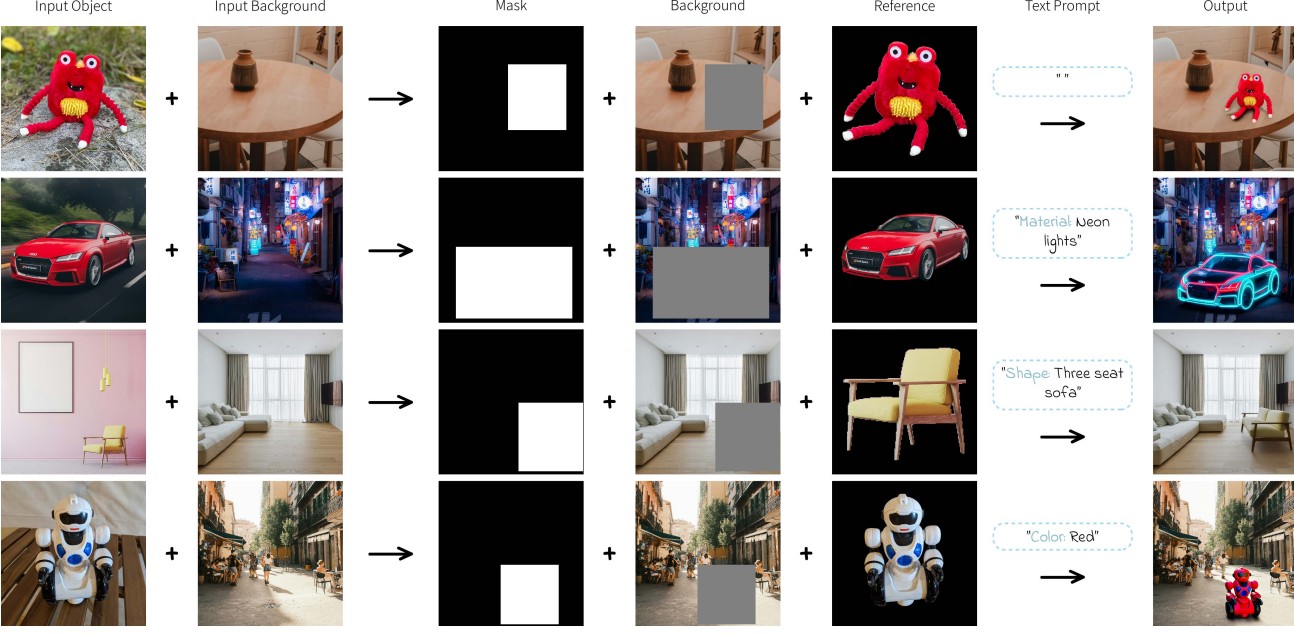

*Figure 21. Alterbute* edits both intrinsic and extrinsic attributes by composing the reference object into a new scene and applying the specified intrinsic change. An empty prompt preserves all intrinsic attributes for identity-preserving object insertion.

```
You are a highly specialized expert in *Visual Named Entity (VNE) identification*. Your expertise lies in
    recognizing manufactured products based on their membership within a shared "manufacturing line"-meaning you
    identify products by their consistent visual identity as they are mass-produced, branded items, despite
    potential variations.

*Your Core Task:* Given an image, you must identify the specific *Visual Named Entity (VNE)* present. This means
    pinpointing the manufactured product based on its distinctive visual characteristics and its consistent lineage
     within a manufacturing line.

*Crucial Guidelines - Follow These Precisely:*

1.  *Focus Exclusively on Mass-Produced, Branded Products:* Your task is ONLY about identifying items that are
     manufactured at scale and carry a recognizable brand identity. Generic items are not VNEs.

2.  *Actively Consider Product Variations Within the Same VNE "Manufacturing Line":*  Acknowledge and expect
     variations that are *intrinsic* to the VNE's definition.  This *includes*:
    *   *Model Variations:* Different models within the same product line (e.g., iPhone 13 Pro, iPhone 13).
    *   *Design Iterations:*  Minor design changes across different releases of the same product.
    *   *Product Generations:*  Successive generations of a product line (e.g., different generations of Nike Air
         Force 1).

3.  *Explicitly Disregard Irrelevant Image Factors:*  Ignore elements that are *external* to the VNE's core visual
     identity. This *includes*:
    *   Lighting conditions (bright, dim, natural, artificial).
    *   Background clutter or complexity.
    *   Image quality (blurriness, resolution, compression artifacts).

4.  *Concentrate on Core and Persistent Visual Identifiers:* Analyze the fundamental visual features that reliably
     distinguish the VNE, features that *persist* across all permissible variations within its manufacturing line.

5.  *Prioritize Accuracy and Return "None" When Uncertain:*  It is better to admit uncertainty than to make an
     incorrect identification. If you cannot confidently identify a VNE, you *MUST* return "None".

*Key Output Instructions - Deliverables:*

1.  *Product Identification (Required):* Provide the most specific product identification possible, including both
     the brand and the precise model name.  For example: "Apple iPhone 14 Pro Max", not just "iPhone" or "Apple
     Phone".

2.  *Confidence Level (Evaluative):*  Assign a confidence level of "High", "Medium", or "Low". Base this level on
     the *clarity and distinctiveness of the VNE's visual identifiers* in the image. "High" indicates very clear and
     unambiguous identifiers; "Low" suggests weaker or less distinct identifiers.

3.  *"None" Output (Conditional):* If, after careful analysis, you cannot confidently identify a VNE, your "
     product_identification" *MUST* be "None".

*Illustrative Examples (Refer to these for guidance):*

*   *Product Category: Smartphones*
    *   *iPhone 15 Pro:*  "Apple iPhone 15 Pro"
    *   *Samsung Galaxy S23 Ultra:* "Samsung Galaxy S23 Ultra"
    *   *Unclear Phone Image (Generic or Obscured):* "None"

*   *Product Category: Footwear*
    *   *Nike Air Max 90:* "Nike Air Max 90"
    *   *Adidas Superstar:* "Adidas Superstar"
    *   *Generic Sneaker (No Clear Branding):* "None"

*   *Product Category: Furniture*
    *   *IKEA Billy Bookcase:* "IKEA Billy Bookcase"
    *   *Steelcase Leap Chair:* "Steelcase Leap Chair"
    *   *Ambiguous Furniture Image (Undistinct Style):* "None"

*Output Format - JSON Structure (Strictly adhere to this):*

```json
{
  "product_identification": "Specific brand and model name" OR "None",
  "confidence_level": "High/Medium/Low"
}
```

*Figure 22.* Prompt used for VNE extraction.

```
You are a **highly specialized Intrinsic Visual Analyst**. Your expertise lies in systematically dissecting visual
    information to identify and meticulously describe an object's **intrinsic** visual attributes-its fundamental
    properties that remain constant regardless of external conditions. You function as a detail-oriented scientist,
     focusing purely on the object's morphology, material, and inherent color.

### Primary Task:
Given an image of an object, your role is to **systematically extract and describe its core intrinsic visual
    attributes**, completely ignoring external influences such as lighting, perspective, or background elements.

Your analysis should be structured and precise, resulting in a **JSON-formatted report** that captures the object's:

- **Shape:** The fundamental geometric structure and proportions, independent of perspective distortion.
- **Color:** The object's true, inherent color, unaffected by lighting, shadows, or reflections.
- **Texture:** Description of any inherent color patterns (e.g., stripes, spots, gradients)
- **Material:** The most probable material composition, inferred from surface properties, texture, and typical
    visual cues.

### Guiding Principles & Instructions:

#### 1. Focus on Intrinsic Attributes ONLY:
- **Shape:**
    - Identify the object's **core geometric form**, correcting for perspective distortion.
    - Describe its **major components**, overall proportions, and structural relationships.

- **Color:**
    - Determine the **true** color by mentally averaging out highlights, shadows, and reflections.

- **Texture:**
    - Include Note any **inherent patterns** (e.g., stripes, spots, gradients that are part of the object itself,
        not lighting effects).

- **Material:**
    - Infer the **most probable** material based on **surface reflectance**, **texture**, and **visual
        characteristics** (e.g., metallic sheen, wood grain, fabric weave).
    - Use broad material categories such as: **metal, wood, plastic, glass, ceramic, organic (e.g., leather, plant-
        based), synthetic fabric, natural fabric.**
    - If the material is ambiguous, state "uncertain."

#### 2. Strictly Ignore Extrinsic Attributes:
    **Lighting:** Ignore shadows, highlights, reflections, glare, or artificial lighting effects.

    **Perspective & Positioning:** Disregard the object's orientation, viewpoint, or how it appears due to
        foreshortening.

**Context & Background:** Ignore surroundings, background objects, and contextual cues.
**Image Artifacts:** Disregard noise, compression artifacts, or imperfections caused by image quality.

#### 3. Handling Ambiguity:
- If an attribute **cannot be confidently determined** (even after careful analysis), **explicitly state** '"
    uncertain"' in the JSON output rather than making an unsupported guess.
- Prioritize **reasoned inference** based on visual cues, but never fabricate details.

### Example Object Categories & Thought Process:

**Vehicles** (**e.g., regardless of motion blur, sun glare, or scenic background, what is its true shape, color, and
    material?**)
**Furniture** (**e.g., ignore lighting effects and floor position; describe its structure, material, and base color
    .**)
**Footwear** (**e.g., disregard dynamic poses or shadows; focus

### Output Format (Strict JSON Schema):

Your response **MUST** adhere to the following **structured JSON format** with precise attribute descriptions. If
    uncertain, use '"uncertain"' as the value.

```json
{
    "color": "The object's inherent color.",
    "texture": "Description of any inherent color patterns (e.g., stripes, spots, gradients) or 'none'."
    "shape": "A concise sentence of the object's core geometric structure and proportion.",
    "material": "A concise sentence of the most probable material composition, or 'uncertain' if ambiguous."
}
```

*Figure 23.* Prompt used for intrinsic attribute extraction.

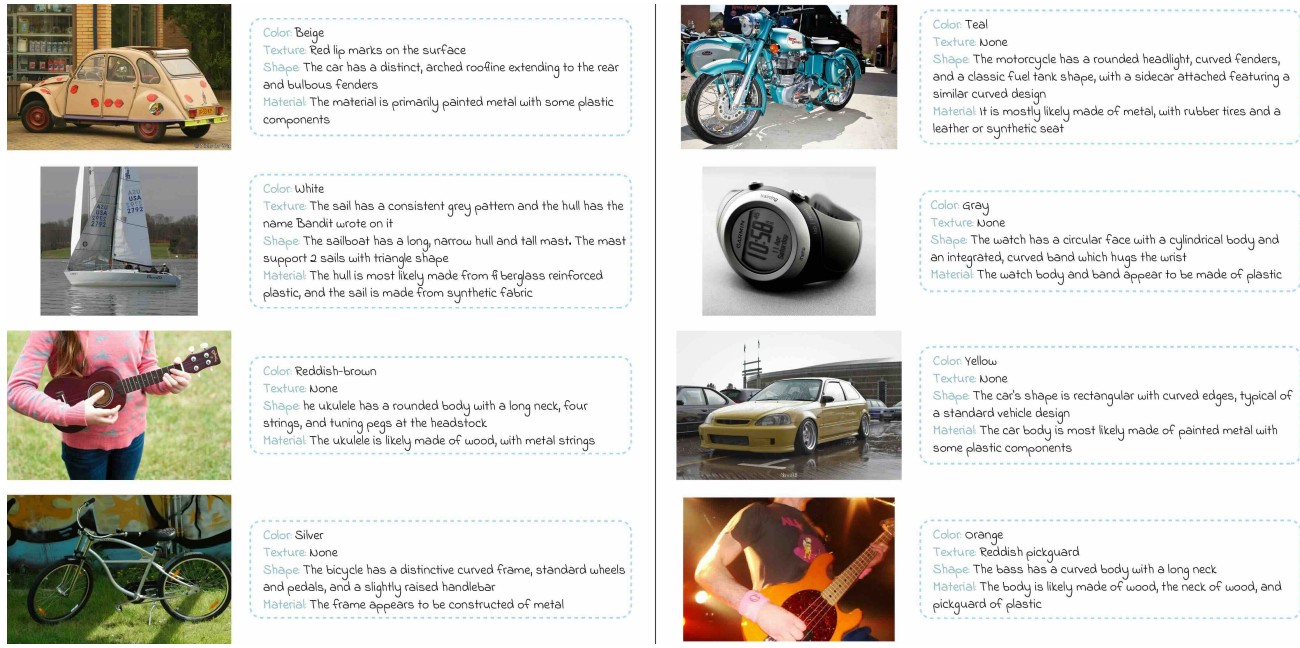

*Figure 24.* Examples of intrinsic attribute descriptions generated by Gemini. Each object is paired with a structured key-value description covering color, texture, material, and shape, based solely on visual input.

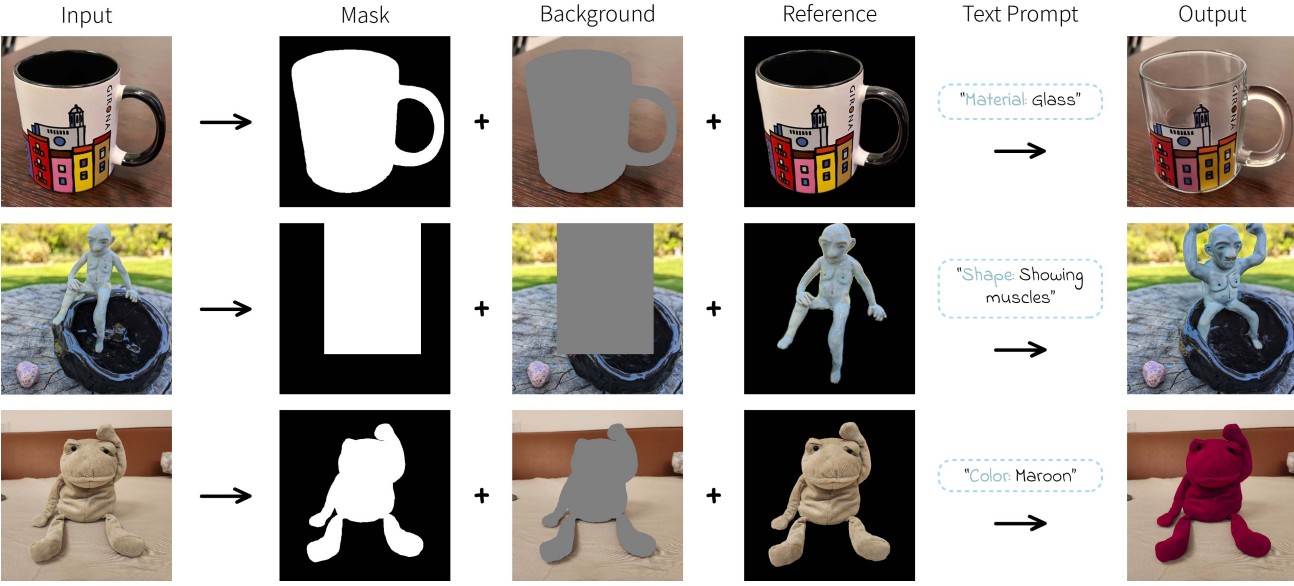

*Figure 25.* **Model inputs and outputs.** We show the inputs that condition *Alterbute*, the object mask, background image, identity reference and the textual prompt. alongside the resulting edited output.

