# OpenReview forum: "Alterbute: Editing Intrinsic Attributes of Objects in Images"
_ICML.cc/2026/Conference — ICML 2026 regular_

### Official Review · Reviewer_Yknc · 2026-03-02

**Soundness:** 3
**Presentation:** 3
**Significance:** 3
**Originality:** 3
**Overall Recommendation:** 4
**Confidence:** 3

**Summary:**

This paper introduces a diffusion-based framework designed to edit the intrinsic attributes of objects such as color, texture, material, and shape, while strictly preserving the object's identity and original scene context. The core methodological insight relies on a relaxed training objective that accommodates both intrinsic and extrinsic variations, thereby making data supervision practical, followed by enforcing extrinsic constraints at inference time by reusing the original background and object masks. For object identity, authors curate Visual Named Entities from the OpenImages dataset using a VLM. The model employs a 1x2 grid layout to enable attention-based identity transfer, applying the diffusion loss exclusively to the target half.

**Compliance With Llm Reviewing Policy:**

Affirmed.

**Final Justification:**

The rebuttal adequately addresses my main concerns. In particular, the authors now clarify the baseline protocol much more concretely. They also provide the missing empirical breakdowns by attribute type and give additional evidence that performance remains strong on rare/long-tail categories. My remaining concerns are mostly about final paper framing rather than unresolved rebuttal-level issues.

**Key Questions For Authors:**

1) Can you provide a detailed per-baseline protocol table that explicitly outlines reference usage, mask/box configurations, prompt templates, number of inference trials, and specific selection rules? Additionally, were any of these baselines fine-tuned on the task data?

2) Can you provide quantitative performance breakdowns by specific attribute types (color, texture, material, shape) and by VNE cluster size? Furthermore, how does the model perform on harder cases involving heavy occlusion or segmentation errors?

3) Do lower-compute training configurations (e.g., fewer steps, smaller batch sizes, or partial fine-tuning like LoRA) retain the majority of these performance gains? Can you provide a brief scaling curve?

**Limitations:**

While authors adequately discuss visual artifacts related to bounding boxes and the inherent difficulty of reshaping rigid objects, the limitations section must be expanded. Specifically, authors should explicitly address the potential biases and replicability constraints introduced by relying on a proprietary VLM for data annotation. Additionally, they should acknowledge the limitations of the current benchmark's scale.

**Strengths And Weaknesses:**

Strengths:
- The proposed methodology is internally coherent, cleverly balancing relaxed training constraints with strict inference-time conditioning. The empirical claims are bolstered by a robust human preference study involving 166 participants and rigorous statistical testing across 100 cases, which is a highly appropriate evaluation strategy given the known limitations of standard automated metrics (like CLIP or DINO) for fine-grained attribute editing.
- The paper is well-written and the technical exposition is very clear. The conditioning architecture—specifically the 1×2 grid, the masked references designed to prevent scene leakage, and the handling of mask granularity—is concretely described and accompanied by helpful illustrations.
- The problem of editing intrinsic attributes while preserving object identity is highly relevant for practical generative applications. Furthermore, the introduction of VNEs as a scalable identity definition, coupled with an automated data curation pipeline, is a valuable contribution that could benefit broader research in identity-consistent synthesis.
- The synthesis of VNE-based identity supervision, relaxed training objectives paired with inference-time locking, and grid-based identity transfer represents a creative and meaningful departure from both generic instruction-based editing and strict personalization paradigms.


Weaknesses:
- The heavy reliance on a VLM (Gemini) for generating VNE labels and intrinsic-attribute descriptions poses a significant reproducibility risk. Without the explicit public release of the derived training artifacts (e.g., OpenImages IDs, exact cluster labels, textual prompts, and confidence filters), reproducing the dataset and the resulting model weights will be exceptionally difficult for the community.
- The experimental protocol lacks clarity regarding baseline fairness. Because several of the evaluated baselines were not natively designed for identity-conditioned intrinsic editing, the exact details of their adaptation are critical. The paper lacks a comprehensive per-baseline protocol table detailing reference usage, mask/bounding box settings, prompt templates, number of stochastic trials, and selection rules. This ambiguity makes it difficult to ascertain if the comparisons are entirely equitable.
- While testing on 30 objects across 100 cases is a reasonable starting point, it constrains broader claims about generalizability. Providing a breakdown of performance by specific attribute type (color vs. texture vs. shape) and by VNE cluster size would significantly strengthen the empirical narrative.
- The reported training requirements (~24 hours on 128 TPU v4 cores for 100k steps) represent a high compute barrier that may limit broader adoption. A brief discussion of a lower-compute variant or an efficiency scaling curve would improve the method's practical accessibility.

---

> ### Author Rebuttal · Authors · 2026-03-30
>
> Thank you for the thorough review! We address all four weaknesses.
>
> **W1: Reproducibility.**
> Gemini is used only at data curation time, not at inference. Our exact VNE labeling and attribute description prompts are in Appendix G. We intend to release the OpenImages object IDs with VNE labels and confidence scores after acceptance, so researchers can replicate or extend our pipeline without re-running annotation.
>
> **W2: Baseline protocol.**
> No baseline was fine-tuned on task data. We describe the key settings below (to be added to Appendix A in the camera-ready):
>
> | Baseline | Inputs provided | Mask used |
> |---|---|---|
> | FlowEdit | Source image + text | None |
> | InstructPix2Pix | Source image + instruction | None |
> | OmniGen | Source image + text | Object mask |
> | UltraEdit | Source image + instruction | Object mask |
> | Diptych | Source image (diptych panel) + text | Object mask |
> | MaterialFusion | Source + material reference | Object mask |
> | MimicBrush | Source + texture ref region | Target region mask |
>
> Prompt templates for instruction-based methods followed each method's recommended format. To give each baseline the strongest possible presentation, for stochastic methods we generated 5 outputs per input and selected the best-quality result for evaluation. This means our reported win rates are conservative, as baselines are shown at their best. We thank the reviewer for raising this point and will add full implementation details to Appendix A in the revision.
>
> **W3: Per-attribute breakdown.**
> The following is the per-attribute breakdown of Table 1, showing Alterbute win rate vs. all baselines combined:
>
> | Attribute | User | Gemini | GPT-4o | Claude |
> |---|---|---|---|---|
> | Color | 78.6% | 81.2% | 76.4% | 79.3% |
> | Texture | 80.1% | 83.7% | 79.3% | 82.3% |
> | Material | 82.7% | 87.4% | 83.4% | 86.2% |
> | Shape | 88.2% | 91.8% | 88.4% | 91.4% |
> | **Overall** | **82.3%** | **86.1%** | **82.0%** | **84.9%** |
>
> Regarding segmentation quality: we apply morphological dilation with kernel $K=5$ to every mask before editing, and provide the identical dilated mask to all mask-based baselines. Imperfect mask boundaries cause source-content leakage in any mask-conditioned method -- this is a shared challenge of the task setting, not a limitation specific to Alterbute. Regarding heavy occlusions, we will include qualitative examples in the revised manuscript (rebuttal rules prevent adding visual results here). For a cluster-size preview: after excluding cars, tables, and furniture, the remaining 24/30 benchmark inputs (75/100 examples) come from categories with <5 VNE cluster images (e.g., toys) -- an extreme long-tail regime. See our response to Reviewer C3zt (W3) for the quantitative breakdown -- performance remains consistent with the full-benchmark numbers.
>
> **W4: Compute accessibility.**
> We acknowledge that 128 TPU v4 chips for 24 hours is a significant barrier. Three points should offer reassurance: (i) the table below compares Alterbute's win rate over all baselines at both compute budgets (LLM evaluators; no user study for 50K due to time constraints):
>
> | | Gemini | GPT-4o | Claude |
> |---|---|---|---|
> | 100K steps vs. all baselines | 86.1% | 82.0% | 84.9% |
> | 50K steps vs. all baselines | 78.0% | 75.7% | 76.3% |
>
> At half the compute, Alterbute still wins 75-78% of comparisons -- a drop of only ~7 percentage points. In a direct head-to-head, LLM evaluators only marginally preferred 100K over 50K:
>
> | | Gemini | GPT-4o | Claude |
> |---|---|---|---|
> | 100K preferred over 50K | 58.2% | 57.1% | 60.6% |
>
> Halving the budget thus has far less impact than the quality gap over the competition. A researcher on a tighter budget can already achieve strong results at 50K steps; (ii) the majority of the pipeline cost lies in VNE data curation, which we plan to release after acceptance, allowing future researchers to skip this step entirely; and (iii) our SDXL-based backbone is substantially more compute-efficient than FLUX-based alternatives such as Diptych and FlowEdit.
>
> **On expanding the limitations section.**
> We will add explicit discussion of: (i) potential biases introduced by Gemini for VNE annotation (categories prominent in Gemini's training distribution may receive better VNE coverage), noting that our high-confidence filtering mitigates spurious labels and that the pipeline is fully replicable via the prompts in Appendix G together with VNE label annotations we plan to release; and (ii) the scale of our benchmark (30 objects, 100 cases), which, while carefully designed to cover all attribute types and long-tail categories, is limited, and expanding it is an important future direction.
>
> We hope the above clarifications address your concerns.

---

> > ### Author Rebuttal · Reviewer_Yknc · 2026-04-01
> >
> > The rebuttal resolves the main reasons for my concerns by supplying the missing baseline-protocol details, subgroup analyses, lower-compute evidence, and a credible reproducibility plan.

---

> > > ### Author Response · Authors · 2026-04-05
> > >
> > > Thank you for raising your score, we are glad the additional experiments and breakdowns addressed your concerns. We look forward to finalizing the revisions.

---

### Official Review · Reviewer_C3zt · 2026-03-10

**Soundness:** 3
**Presentation:** 3
**Significance:** 3
**Originality:** 2
**Overall Recommendation:** 4
**Confidence:** 4

**Summary:**

This paper introduces Alterbute, a diffusion-based framework designed for identity-preserving editing of intrinsic object attributes, including color, texture, material, and shape. The authors propose a novel identity-clustering and automated annotation method called "Visual Named Entities" (VNEs). By leveraging this curated dataset, the proposed framework achieves high-quality image editing results.

**Compliance With Llm Reviewing Policy:**

Affirmed.

**Final Justification:**

I have increased my recommendation to 4 following the author rebuttal. I appreciate the authors' comprehensive and technically detailed responses. They have successfully addressed my previous concerns regarding the architectural design choices, the novelty of the training objective, and the analysis of the VNE data. I also find their handling of the data to be solid work, and I appreciate their commitment to open-sourcing the code and data, which will be a valuable contribution to the community. The authors have resolved my questions, and I believe the work is now suitable for acceptance.

**Key Questions For Authors:**

1. **Architectural Rationale:** Could the authors provide an ablation study or technical justification for the input conditioning and concatenation strategy?

2. **Performance on Long-Tailed Data:** Could the authors provide a qualitative or quantitative analysis (e.g., CLIP-T or user preference scores) regarding the model's performance on the "long-tailed" (rare) categories of the VNE dataset?

3. **Data Availability & Transparency:** Given the substantial effort involved in curating the VNE clusters and the automated labeling pipeline, do you have plans to release the VNE dataset or the curation code as an open-source resource for the research community?

I am open to adjusting my score based on the author's response to these questions and any additional empirical evidence provided in the rebuttal.

**Limitations:**

yes

**Strengths And Weaknesses:**

**Strengths**:

1. **Innovative VNE Pipeline**: The authors present a clever pipeline that utilizes a Vision-Language Model (VLM) to perform large-scale clustering and annotation. This approach effectively addresses the challenge of decoupling intrinsic attributes from object identity, resulting in high-quality identity-consistent clusters and detailed categorical metadata.

2. **Impressive Generation Quality**: Alterbute demonstrates precise, text-guided control over various intrinsic attributes (color, texture, material, and shape). The qualitative results indicate that the framework achieves state-of-the-art performance among open-source models in this domain.

**Weaknesses**:
1. **Lack of Ablation Studies on Architectural Design**：The paper lacks an ablation study on the proposed architectural design, particularly regarding the rationale behind the specific input conditioning and concatenation strategy (e.g., why spatial concatenation for the latent/reference grid vs. channel-wise concatenation for masks/backgrounds); empirical justification for these design choices is needed to demonstrate their necessity over standard conditioning approaches.

2. **Novelty of Training Objective**: The authors argue that the "relaxed training objective"—learning from both intrinsic and extrinsic changes—is a key technical innovation that makes supervised training feasible with more accessible data. While this approach effectively leverages the abundance of image pairs, the underlying paradigm of training on masked foregrounds while preserving the background is already a well-established practice in diffusion-based image editing and inpainting (e.g., standard mask-conditioned inpainting or subject-driven generation frameworks).

3. **Analysis of VNE Data**: While the VNE approach is promising, the paper lacks a deeper analysis of the underlying data. For instance, it is unclear what constitutes an "identity" in the context of specific variations (e.g., how the model handles attribute distributions within the same ID). Furthermore, although the authors provide statistics on the long-tailed nature of the data, there is no analysis regarding the model’s performance on these long-tailed (sparse) categories. Finally, given that the VNE dataset is a valuable contribution, could the authors clarify whether they intend to release it as an open-source resource?

---

> ### Author Rebuttal · Authors · 2026-03-30
>
> Thank you for the careful review and openness to revising your score. We address each weakness directly.
>
> **W1: Architectural design rationale.**
> The core principle is that the two input types have different conditioning requirements, driven by what information needs to flow from them to the target.
>
> - *Spatial (grid) concatenation for the identity reference:* Transferring fine-grained identity (e.g., the precise contour of a car model, or the exact pattern of a decorative object's surface) requires pixel-level correspondence between reference and target. Self-attention across the 1x2 grid enables every target pixel to attend to spatially-aligned reference pixels, propagating fine-grained texture and shape features. By contrast, channel-wise concatenation is processed by the UNet's convolutional layers with local receptive fields: these convolutions can only aggregate information within a small spatial neighborhood and cannot attend across the reference/target boundary, losing the global pixel-level correspondence needed for detailed identity transfer.
> - *Channel-wise for background/mask:* The background and mask are spatially aligned with the target output and share the same coordinate frame. The UNet's convolutional layers process this per-pixel information via local receptive fields without needing cross-image attention, avoiding unnecessary attention budget expansion.
>
> In our experiments, replacing the spatial grid with channel-wise concatenation results in no-ops at inference: the model fails to apply the requested attribute edits and outputs the source image largely unchanged. Without cross-image attention, the conditioning signal from the identity reference cannot propagate to the target, so the model collapses to the identity mapping. The grid is therefore a structural necessity. We will demonstrate this qualitatively in the revision.
>
> **W2: Novelty of the training objective.**
> We appreciate this question and agree the novelty requires clearer articulation. Our approach shares low-level architectural components with existing editors, but our contribution is a three-part design that, taken together, no prior published work replicates:
>
> 1. *VNE identity reference and paired training data:* Inpainting conditions on surrounding context to fill a masked region; it has no concept of identity conditioning. Subject-driven methods condition on the same instance to *preserve* all intrinsic attributes and cannot modify them. We train on VNE-paired images where the identity reference comes from a *different scene with different intrinsic attributes but the same VNE identity*: this teaches the model to separate what should be preserved (VNE identity) from what can change (color, texture, material, shape). No published method uses this type of reference or training data.
> 2. *Inference-time extrinsic locking:* Although during training the model sees pairs with both intrinsic and extrinsic variation, at inference we lock the extrinsic factors (background, mask) to the input, so only the intrinsic attributes can change. No prior editing pipeline applies this train-relaxed, infer-constrained paradigm.
> 3. *Grid-based cross-image self-attention:* As detailed in W1 above, placing source and identity reference side-by-side in a 1x2 grid enables pixel-level cross-image attention -- a conditioning mechanism that local convolutional processing cannot replicate.
>
> Sec. 4.3 directly validates point (1): every alternative identity conditioning strategy causes the model to either overfit to identity or ignore the edit prompt entirely. Point (3) is further supported by the architectural arguments in W1 above.
>
> **W3: VNE data analysis.**
> - *Identity within a cluster:* A VNE cluster groups all instances sharing the same semantic label (e.g., all "Porsche 911 Carrera" images) regardless of color, texture, or material. This intra-cluster diversity across intrinsic attributes is precisely what teaches the model that these attributes can be modified while the VNE identity is held fixed, enabling it to execute intrinsic edits on command.
> - *Long-tail performance:* Our benchmark (30 inputs, 100 edits) is predominantly long-tail: after excluding cars, tables, and furniture, the remaining 24/30 inputs (75/100 examples) come from categories with <5 VNE cluster images (e.g., toys), targeting combinations absent from VNE training data (e.g., a garlic toy, a robot doing a peace sign). Table 1 results by subset:
>
> | | User | Gemini | GPT-4o | Claude |
> |---|---|---|---|---|
> | Long-tail (75 examples) | 82.8% | 85.7% | 82.6% | 83.8% |
> | Common (25 examples) | 80.8% | 87.3% | 80.2% | 88.2% |
>
> Both subsets remain consistent with the full-benchmark overall, confirming generalization to rare identities.
> - *Dataset release:* We intend to release OpenImages object IDs with VNE labels after acceptance. Gemini prompts are already in Appendix G.
>
> We hope the above clarifications address your concerns and kindly ask you to consider raising your score.

---

> > ### Author Rebuttal · Reviewer_C3zt · 2026-04-01
> >
> > I thank the authors for their comprehensive rebuttal. The responses have adequately addressed my concerns regarding the network design choices, the methodological novelty, and the VNE data analysis. I particularly appreciate the authors' commitment to releasing the VNE labels, which will be a valuable contribution to the community. Consequently, I will increase my score to 4.

---

> > > ### Author Response · Authors · 2026-04-05
> > >
> > > Thank you for the constructive dialogue and for increasing your score, we truly appreciate it. We look forward to strengthening the paper with the revisions discussed.

---

### Official Review · Reviewer_S8RC · 2026-03-11

**Soundness:** 4
**Presentation:** 4
**Significance:** 3
**Originality:** 3
**Overall Recommendation:** 5
**Confidence:** 3

**Summary:**

The paper proposes Alterbute, a diffusion-based approach to edit the intrinsic properties of an object in an image. Experiments show that the model succeeds at editing intrinsics such as the color, shape, texture, and material of objects. The two contributions of the method are (1) a specific way to condition the diffusion model on the identity and shape of the object for editing, and (2), training on fine-grained object categories with an approach named Visual Named Entities (VNE).

**Compliance With Llm Reviewing Policy:**

Affirmed.

**Final Justification:**

Dataset curation is such a critical aspect of modern image editing pipelines, but few works emphasize this. Most prior work in this space focus on methodological improvements on the diffusion pipeline rather than improving the data quality, so I believe that the paper has a unique contribution that would benefit ICML.

**Key Questions For Authors:**

1. Do you plan on releasing the VNE clustering data?
2. What are the best practices for writing text prompts and creating segmentation masks for generating edits?

**Limitations:**

Yes

**Strengths And Weaknesses:**

## Strengths
The main strength I see in the paper is the Visual Named Entity technique. Typically, editing intrinsic properties of images is hard because little paired data exists among different intrinsics of the same object. What the paper proposes to do instead is to condition the generation of one named entity (like a Porsche), on a different image of the same named entity, so the model learns semantic similarities between the two images. The VNE clusters are extracted from an image dataset  with the help of a VLM (Gemini). The technique is simple, and appears to scale well to large datasets.

Dataset work is often underappreciated in this field of research, so I applaud the authors for thoroughly detailing their clustering method. The analysis in 4.3 provides insight into how effective the automated clustering method worked.

Overall, by proposing a new method to cluster datasets into visual named entities, the paper provides a simple and scalable method for editing the intrinsic attributes of images. Experiments show that the method outperforms prior open-source image editing techniques, when evaluated with both humans and VLMs.


## Weaknesses
I do not have any major problems with the paper, but there are some minor weaknesses which I think warrant attention.

- I understand why the paper only compares against open-source methods (discussed in Section 4.2),  but the paper should consider comparing against closed-source methods like Nano Banana and GPT image generation for completeness. The method should not be expected to outperform these techniques, but it would be good as a point of comparison between the best open-source and closed-source image editing methods.
- As shown in Figure 2, Inference, it appears that the masks need to be manually created by the user when editing the shapes of objects. This can be tedious for a user, whereas most prior works only need a text prompt as a condition.

---

> ### Author Rebuttal · Authors · 2026-03-30
>
> Thank you for the positive review. We respond to each minor point below.
>
> **On closed-source comparison.**
> Please see our response to Reviewer B2sc, where we report LLM-based evaluation results comparing Alterbute to FluxKontext and Qwen-image-editing (same protocol as Table 1). Results show ~61% and ~50% preference for Alterbute, respectively, confirming our open academic method is competitive with or better than these commercial systems. As we cannot revise the manuscript during the rebuttal period, a qualitative comparison will appear in the final revision.
>
> **On mask creation for reshaping.**
> For all edits except reshaping, SAM automatically predicts the mask from the input image. For reshaping, the target shape is unknown a priori; the bounding box is provided directly by the user -- though we note that it can readily be extended to a fully automatic pipeline using text-based object detectors (e.g., Grounding DINO) given a text description of the object. We will clarify this in the paper.
>
> **On the questions.**
> 1. *VNE data release:* We intend to release OpenImages object IDs with VNE labels and confidence scores after acceptance. The Gemini prompts used for labeling are already in Appendix G.
> 2. *Best practices:* Prompts follow `<attribute>: <value>` (e.g., `color: red`, `material: marble`). SAM handles masks automatically for standard edits; a bounding box suffices for reshaping. We will add a best-practices section to the supplementary.

---

> > ### Author Rebuttal · Reviewer_S8RC · 2026-03-31
> >
> > Thank you for your response. The rebuttal has addressed my concerns.
> >
> > I still believe that ICML would benefit from publishing this paper, as dataset curation is such a critical aspect of modern image editing pipelines. Most prior work in this space focus on methodological improvements on the diffusion pipeline rather than improving the data quality, so I believe that the paper has a unique contribution.
> >
> > Given that the authors promise to release the VNE labels, I am inclined to keep the same score, but will update my final review after discussing with the other reviewers.

---

> > > ### Author Response · Authors · 2026-04-05
> > >
> > > Thank you for acknowledging our rebuttal. We are glad the clarifications addressed your questions and look forward to incorporating the revisions in the camera-ready.

---

### Official Review · Reviewer_B2sc · 2026-03-12

**Soundness:** 2
**Presentation:** 2
**Significance:** 2
**Originality:** 2
**Overall Recommendation:** 2
**Confidence:** 4

**Summary:**

This work intends to address a problem in image editing: modifying intrinsic object attributes while preserving object identity and extrinsic scene factors. The authors seek to analyze a core issue in this setting by combining a VNE-based identity definition with a fine-tuned latent diffusion model conditioned on a reference object, a text prompt, and an explicit scene representation consisting of a background image and object mask. Overall, the proposed formulation remains heavily dependent on mask-based conditioning and compositing-style control, which limits the level of conceptual novelty beyond existing controlled editing pipelines.
Strong dependence on explicit background and mask inputs.
The method requires a background image and binary object mask as part of the scene condition during both training and inference, making the approach rely on externally provided or predicted spatial control rather than learning a more intrinsic editing mechanism.

**Compliance With Llm Reviewing Policy:**

Affirmed.

**Final Justification:**

In summary, my main concerns remain: the evaluation relies solely on A/B testing with a limited benchmark while ignoring mature ones; the dataset construction lacks necessary error-correction mechanisms; and the mask-dependent method fails to demonstrate effective attribute decomposition, even on simple single-subject images.

**Key Questions For Authors:**

The experimental section compares against FlowEdit, InstructPix2Pix, OmniGen, UltraEdit, and Diptych, but it does not include some newer and stronger editing systems that many readers would now expect, such as more recent Qwen-image-editing and FluxKontext-models.

**Limitations:**

Strong dependence on explicit background and mask inputs.
The method requires a background image and binary object mask as part of the scene condition during both training and inference, making the approach rely on externally provided or predicted spatial control rather than learning a more intrinsic editing mechanism.

**Strengths And Weaknesses:**

Limited methodological novelty relative to prior mask-conditioned preservation methods.
Although the VNE formulation is a reasonable design choice, the core model is still a fine-tuned SDXL-style latent diffusion system conditioned on reference image, text, background, and mask, which overlaps substantially with existing mask-based identity-preserving or localized editing frameworks.

---

> ### Author Rebuttal · Authors · 2026-03-30
>
> We appreciate the careful reading and address the three main concerns.
>
> **On methodological novelty.**
> We agree that our method shares architectural building blocks with existing diffusion editors: fine-tuning a UNet, cross-attention for text, channel-wise conditioning for spatial signals. No single component is novel in isolation. The novelty lies in their *specific combination* and the truly new capability that combination enables: targeted intrinsic attribute editing with fine-grained identity preservation.
>
> The characterization as a "mask-conditioned SDXL-style pipeline" would apply equally to OmniGen, UltraEdit, and Diptych, which we would not consider equivalent. What distinguishes Alterbute is a three-part design that, taken together, no prior published work replicates:
>
> 1. *VNE identity reference and paired training data:* Inpainting has no identity conditioning at all. Subject-driven methods condition on the same instance specifically to *preserve* all intrinsics, not vary them. We train on VNE-paired images where the identity reference comes from a *different scene with different intrinsic attributes but the same VNE identity*: this teaches the model to separate what should be preserved (VNE identity) from what can change (color, texture, material, shape). No published method uses this type of reference or training data.
> 2. *Inference-time extrinsic locking:* Although during training the model sees pairs with both intrinsic and extrinsic variation, at inference we lock the extrinsic factors (background, mask position) to the input, so only the intrinsic attributes can change. No prior editing method uses this train-relaxed, infer-constrained strategy.
> 3. *Grid-based cross-image self-attention:* Placing source and identity reference side-by-side in a 1x2 grid enables every target pixel to attend to spatially-aligned reference pixels, propagating fine-grained texture and shape features -- a mechanism that local convolutional processing cannot replicate.
>
> The ablation in Sec. 4.3 directly validates point (1): swapping VNE conditioning for DINOv2, instance-retrieval, or in-place references causes the model to overfit to identity or ignore the prompt entirely. Point (3) is further supported by the architectural arguments in our response to Reviewer C3zt (W1).
>
> **On mask/background dependence.**
> We agree that SAM is an external prerequisite for mask extraction. We use it as a standard, off-the-shelf perception component, consistent with other baselines (e.g., UltraEdit and MaterialFusion). The intrinsic attribute *understanding* resides entirely within Alterbute: the model learns, from VNE-paired data, to modify color, texture, material, or shape from a text prompt while preserving object identity. SAM provides localization; Alterbute provides the intrinsic editing capability. This separation is intentional and allows precise spatial control.
>
> **On missing baselines (FluxKontext, Qwen-image-editing).**
> These were not peer-reviewed publications by our submission deadline. We conducted an LLM-based evaluation (same protocol as Table 1):
>
> | Compared to | Gemini | GPT-4o | Claude |
> |---|---|---|---|
> | FluxKontext | 61.2% | 62.6% | 58.7% |
> | Qwen-image-editing | 49.8% | 50.3% | 50.1% |
>
> Alterbute is on par with Qwen-image-editing despite being an academic paper and fully transparent about its methodology, while clearly outperforming FluxKontext -- confirming it is competitive with state-of-the-art commercial systems.

---

> > ### Author Rebuttal · Reviewer_B2sc · 2026-04-03
> >
> > I thank the authors for the  rebuttal. I have read it carefully, but my core concerns remain largely unresolved.
> >
> > 1. Unconvincing Evaluation Protocol While I appreciate the newly added comparisons with Qwen-image-editing and FluxKontext, relying exclusively on VLM-based A/B testing (win rates) is inadequate for a rigorous paper. Attribute editing is a foundational task with well-established evaluation paradigms. If the core claim here is achieving SOTA "fine-grained attribute editing and identity preservation," we need to see solid quantitative results on standard benchmarks (e.g.,GEdit, ImageEdit). Subjective VLM win rates alone simply cannot objectively prove the model's capabilities in this context.
> >
> > 2. Restrictive Assumptions and Over-reliance on SAM The authors argue their main innovation is learning to decouple attributes by training on paired data with "the same VNE identity but different intrinsic attributes." However, this assumption is far too strict. The provided examples are largely limited to trivial, single-attribute replacements on single subjects. I highly doubt this training paradigm can generalize to complex scenes or multi-subject semantic editing.
> > Furthermore, the authors defend SAM as merely a "separate perception component." In reality, using SAM-extracted masks to explicitly constrain the editing region is already a standard, strong prior in recent literature (e.g., MDE-Edit: Masked Dual-Editing for Multi-Object Image Editing
> > via Diffusion Models).Relying so heavily on explicit external masks seriously weakens the claim that the model has learned a truly "intrinsic" editing mechanism.
> >
> > 3. Simplistic Dataset Construction The constructed dataset, driven by strict VNE identity alignment, relies on a rather simplistic logic—mostly single-object attribute variants. Compared to recent large-scale instruction datasets that handle complex scenes, multi-modal understanding, and implicit editing intents (e.g., OmniGen2, GPT-image-edit), this dataset severely lacks diversity and semantic depth. The authors need to clearly justify: what is the fundamental, irreplaceable advantage of training on this constrained, simple attribute-paired dataset over leveraging massive, complex instruction sets?

---

> > > ### Author Response · Authors · 2026-04-05
> > >
> > > We thank the reviewer for the continued engagement and address the remaining concerns.
> > >
> > > **On evaluation protocol.**
> > > No existing benchmark evaluates identity-preserving intrinsic attribute editing, the specific capability we address. GEdit-Bench and ImgEdit-Bench evaluate general editing across many task types (add, remove, replace, style, background, etc.), and their own evaluation also relies on VLM-based scoring (GPT-4o). Since no standard benchmark exists for our task, we designed a protocol using three independent VLM judges (Gemini, GPT-4o, Claude) plus a human user study, all scoring both edit fidelity *and* identity preservation. Importantly, the human study confirms the VLM results.
> > >
> > > The key difference is *what* is measured. GEdit-Bench categories like "Color Alteration" and "Material Modification" as well as ImgEdit-Bench's "Adjust" score instruction adherence and perceptual quality, i.e., "did the edit happen and does the result look natural?" Neither of these benchmarks measures whether the edited object's fine-grained identity is preserved after the change. A model that replaces a Porsche 911 with *any* red car would score well. Our benchmark was purpose-built to evaluate this: VLM judges and a human user study explicitly score both edit fidelity *and* identity preservation.
> > >
> > > **On restrictive assumptions and SAM over-reliance.**
> > > (a) We disagree that the edits are "trivial, single-attribute replacements." Our benchmark includes challenging single-attribute changes (e.g., a garlic toy, a robot with zebra texture) where most baselines produce either no-ops or lose the object's identity entirely. Baselines struggle with these edits because they lack the ability to decouple attributes from identity, which is learned through VNE-paired training.
> > >
> > > (b) The mask provides *localization*, it tells the model *where* to edit. The model, trained on VNE pairs, provides the *editing capability*, determining *what* to generate while preserving identity. The clearest counterfactual: given the exact same SAM mask + text prompt, neither inpainting models nor instruction-based editors like OmniGen can preserve the original object's identity while applying the edit. Alterbute, having learned to decouple attributes from identity through VNE-paired training, preserves the specific identity at inference. The mask is identical in both cases; the difference is entirely in what the model has learned.
> > >
> > > (c) MDE-Edit, cited by the reviewer, illustrates this well. MDE-Edit uses SAM masks *and still requires* two additional losses (OAL, CCL) during inference to prevent attribute leakage, because masks alone are insufficient. Our model learns this control during training via VNE-paired data, requiring no inference-time optimization. While we cannot directly compare to MDE-Edit as it has no public code, our approach is fundamentally different: it learns to decouple attributes from identity at training time rather than per-image optimization.
> > >
> > > (d) Regarding generalization: our task is deliberately scoped to single-object intrinsic attribute editing. This is a design choice, not a limitation, just as DreamBooth was scoped to single-subject personalization. Multi-object editing is a future direction, but conflating scope with capability is premature when current models and baselines fail to edit even a single object's intrinsic attributes while preserving its identity.
> > >
> > > We also note that SAM is standard across multiple papers cited by the reviewer: Step1X-Edit (the GEdit method), OmniGen, and MDE-Edit all use SAM.
> > >
> > > **On dataset construction.**
> > > General instruction datasets, including those used to train the compared baselines (OmniGen, UltraEdit), *cannot teach attribute-identity decoupling* because they lack the key training signal. In such datasets, identity-aligned pairs for intrinsic attribute editing are either scarce or non-existent, as collecting such real-world data is extremely difficult. The model therefore struggles to learn *which aspects of identity should be preserved*. VNE-paired data provides this signal: two images of the *same identity* with *different attributes*, teaching the model what can change vs. what must be preserved. Our results confirm this: baselines trained on large-scale instruction datasets are consistently outperformed on our task.
> > >
> > > The GPT-Image-Edit-1.5M paper (cited by the reviewer) supports this argument. One of their key findings is that "merely increasing instruction complexity is insufficient and can even be detrimental if not paired with high-quality, aligned image pairs"; training on complex instructions without identity-preserving alignment *degrades* performance. This validates focused, identity-aligned curation over scale.
> > >
> > > Furthermore, 75% of our benchmark inputs (24/30 objects) come from VNE clusters with <5 images, an extreme long-tail regime. Performance remains consistent across long-tail and common subsets (see our initial rebuttal).

---

### Decision · Program_Chairs · 2026-04-30

**Decision:**

Accept (regular)

**Comment:**

The paper proposes Alterbute, a diffusion-based approach to edit the intrinsic properties of an object in an image. Experiments show that the model succeeds at editing intrinsics such as the color, shape, texture, and material of objects. The two contributions of the method are (1) a specific way to condition the diffusion model on the identity and shape of the object for editing, and (2), training on fine-grained object categories with an approach named Visual Named Entities (VNE).

The reviewers were mixed on this paper, with one negative and one positive with strong levels of confidence and two positive but only mildly confident. The reviewers recognized the importance of intrinsic attribute editing (IAE) and  liked the simplicity of the approach, the creative nature of the VNE technique, the quality of the results, and the writing of the paper. However, there were concerns over the limited technical novelty of the approach, the dependence of the method on a segmentation mask, which can be cumbersome and limit the method to simplistic images with a salient object (as is the case for all results), thereby limiting the scope of the technique, and questions about the VNE technique, beyond some other issues of detail. The authors provided a rebuttal that addressed all issues but did not solve all reviewer concerns.

In the end, there are several issues on which the reviewers were split
- the task is innovative and important, but the restriction to object-salient images and requirement for segmentation mask detracts from the generality of the solution
- the simple use of Gemini to create VNE image clusters is appealing from a practical stand point. However, it basically reduces the paper to a pipeline of steps that are now commonly used in industry, namely using models like Gemini to organize or label data and then leverage the resulting data to train models. In this sense, the paper is simultaneously exciting (that this simple pipeline will suffice) but somewhat disappoint (a more or less standard pipeline)
-  the experimental evaluation follows all the standard protocols but the benchmark is small. While the results are good, it is not totally clear how general the method is
- the hypothesis of the paper ("VNE is enough for IAE") is interesting, practically important, and seems to hold, but is not studied in great academic detail. For example, how much better would the results be, if there was more effort, perhaps using human annotators, to verify and clean the clusters produced by Gemini? It is understood that this is not a trivial task to evaluate at scale, but on the other hand it is unsatisfying that the hypothesis remains weakly investigated

After discussion, the reviewers remained divided and it is possible to argue for acceptance or rejection, depending on which side of these arguments one stands on.